# Uncertainty Estimation for 3D Object Detection via Evidential Learning

## Abstract

3D object detection is an essential task for computer vision applications in autonomous vehicles and robotics. However, models often struggle to quantify detection reliability, leading to poor performance on unfamiliar scenes. We introduce a framework for quantifying uncertainty in 3D object detection by leveraging an evidential learning loss on Bird's Eye View representations in the 3D detector. These uncertainty estimates require minimal computational overhead and are generalizable across different architectures. We demonstrate both the efficacy and importance of these uncertainty estimates on identifying out-of-distribution scenes, poorly localized objects, and missing (false negative) detections; our framework consistently improves over baselines by 10-20% on average. Finally, we integrate this suite of tasks into a system where a 3D object detector auto-labels driving scenes and our uncertainty estimates verify label correctness before the labels are used to train a second model. Here, our uncertainty-driven verification results in a 1% improvement in mAP and a 1-2% improvement in NDS.

## 1 Introduction

Detecting 3D objects from LiDAR and multiple camera images is crucial for autonomous driving. Recent techniques mostly rely on bird's eye view (BEV) representations (Zhou & Tuzel, 2018; Lang et al., 2019; Philion & Fidler, 2020; Yin et al., 2021), where information from the different sensors is fused to generate a consistent representation within the ego-vehicle's coordinate system. BEV effectively captures the relative position and size of objects, making it well-suited for perception and planning (Ng et al., 2020; Liang et al., 2022; Chen et al., 2023).

Deep neural networks are excellent at performing detections but assessing their reliability remains a challenge. Sampling-based uncertainty estimation methods, like MC-Dropout (Gal & Ghahramani, 2016) and Deep Ensembles (Lakshminarayanan et al., 2017), are among the most common approaches used for this purpose. MC-Dropout works by randomly deactivating network weights and observing the impact, while Deep Ensembles involve training several networks with different initializations. Although intuitive, these methods typically require a multiplier on the nominal compute, memory, or training costs of the neural network. Consequently, these methods are inviable for large-scale applications including the 3D detection systems used for autonomous vehicles. Moreover for common tasks such as classification or regression, sampling-free uncertainty estimation methods are easier to implement and deploy (Wen et al., 2020; Durasov et al., 2021; 2024; Laurent et al., 2023; Ashukha et al., 2020; Durasov et al., 2022b). In this space, Evidential Deep Learning (EDL) has recently emerged as a promising alternative for providing high-quality epistemic uncertainty estimates at low computational overhead (Sensoy et al., 2018; Amini et al., 2020).

In this paper, we introduce a computationally efficient uncertainty estimation framework for 3D object detection motivated by EDL. The 3D detection task differs from classical uncertainty estimation applications due to the multi-faceted nature of detection uncertainty in class label and localization, as well as the variety of representations of the data. Consequently, we propose a generic approach to measure the uncertainty of estimates in each cell of the BEV representation. We transform the BEV heatmap head found in modern 3D detection models to an EDL-based head and employ a regularized training loss to learn to generate these uncertainties. By aggregating these BEV-level uncertainty estimates, we can simultaneously predict objectness probabilities and uncertainties as-

Figure 1: **3D Object Detection Uncertainty Estimation Framework**. Our Evidential Deep Learning approach jointly generates heatmap probabilities for objects within Bird's Eye View and their corresponding uncertainty values, which allows us to detect several critical problems within autonomous driving, namely **(left)** identifying out-of-distribution scenes (e.g., with bad weather conditions), **(middle)** erroneous predicted boxes, and **(right)** missed objects (e.g., missed grey and white cars in the image). The uncertainty estimates guide selective human verification, leading to improvements in detection metrics (e.g., mean Average Precision (mAP) and nuScenes Detection Score (NDS)).

sociated with class and location. To demonstrate the efficacy of this method, we adapt it to several downstream uncertainty-quantification tasks in autonomous driving, including:

• **Out-of-distribution (OOD) scene detection**: By using EDL-based uncertainty estimates, the model can effectively detect scenes that differ significantly from the training distribution (8% improvement compared to other uncertainty baselines).

• **Bounding-box quality assessment**: By using the provided uncertainty estimates, the model can effectively assess the quality of predicted bounding boxes, enabling more reliable predictions (7% improvement compared to other uncertainty baselines).

• **Missed objects detection**: By leveraging uncertainty estimation, the model can effectively highlight regions where objects are potentially missed, addressing the critical issue of false negatives (5% improvement compared to other uncertainty baselines).

Finally, we integrate these experiments into a unified pipeline where a 3D object detector automatically labels driving scenes. Using uncertainty estimates, we identify which outputs — at the scene, bounding box, or missed object level — need human verification. This focused verification step enhances the performance of the secondary model, resulting in significant improvements in final detection metrics through uncertainty-driven refinement. Our approach is illustrated in Fig. 1.

## 2 RELATED WORK

### 2.1 3D OBJECT DETECTION

3D object detection has gained significant attention in recent years driven by advances in deep learning and large-scale datasets. This task is broadly classified into three main approaches (Chen et al., 2023): camera-based, LiDAR-based, and multi-modal approaches. Recent camera-based methods predict 3D object from multi-view camera images (Wang et al., 2023; Li et al., 2022; Liu et al., 2022). This method aggregates features from multiple camera views to construct a comprehensive understanding of geometry. LiDAR-based methods estimate 3D objects in given point clouds. This method projects point clouds onto a regular grid such as pillars (Lang et al., 2019), voxels (Zhou & Tuzel, 2018), or range images (Fan et al., 2021) because of irregular nature of point clouds, and then deep model are used to get features for object detection. Recent advancements have explored the integration of camera with LiDAR to further enhance 3D detection capabilities (Bai et al., 2022; Liang et al., 2022; Wang et al., 2021). This multi-modal strategy enables the model to leverage the complementary strengths of each sensor, yielding improved detection accuracy over single modality methods (Bai et al., 2022). However, despite their development and effectiveness, these models

struggle to adequately assess their own confidence in the predictions they make. It therefore does not know how uncertainty or reliable of estimation is.

## 2.2 UNCERTAINTY ESTIMATION

Uncertainty estimation is the study of quantifying the reliability of a model's predictions. Most uncertainty estimation methods must balance the quality of estimated uncertainties with the computational cost of generating such estimates. Deep Ensembles, which train multiple neural networks with different initializations, provide more reliable uncertainty estimates than most alternatives (Lakshminarayanan et al., 2017; Ovadia et al., 2019; Antorán et al., 2020; Gustafsson et al., 2020; Ashukha et al., 2020; Daxberger et al., 2021; Postels et al., 2022). However, their high compute and memory demands make them impractical for large-scale 3D detection networks. MC-Dropout is a widely used method for generating uncertainty estimates with low computational cost by randomly deactivating network weights (Gal & Ghahramani, 2016). However, its estimates are generally less reliable than those of Deep Ensembles (Ashukha et al., 2020; Wen et al., 2020; Durasov et al., 2021). In between these two approaches lie a broad family of Bayesian Networks (Mackay, 1995) that achieve varying performances by trading off compute (Blundell et al., 2015; Graves, 2011; Hernández-Lobato & Adams, 2015; Kingma et al., 2015). Nonetheless, most of these uncertainty estimation methods require multiple forward passes during inference, making them impractical for fast inference. As a result, sampling-free single-pass approaches will modify the network architecture or training process to generate fast estimates, albeit for task-specific settings (Choi et al., 2021; Postels et al., 2019; Malinin & Gales, 2018; Amersfoort et al., 2020; Malinin & Gales, 2018; Mukhoti et al., 2021; Durasov et al., 2022a; Ashukha et al., 2020).

EDL is an increasingly popular single-pass paradigm for uncertainty estimation that use a single neural network to estimate a meta-distribution representing the uncertainty over the predicted distribution (Sensoy et al., 2018; Amini et al., 2020; Ulmer et al., 2021). EDL methods are more computationally efficient than most Bayesian and ensemble methods, while requiring only minor architectural changes to the network. Most importantly, these methods have demonstrated strong performance in a variety of applications involving uncertainty quantification and in detecting out-of-distribution (OOD) or novel data (Bao et al., 2021; Aguilar et al., 2023; Zhao et al., 2023).

## 3 METHOD

EDL adapts the model architecture and loss function of a nominal learning problem to generate uncertainty estimates along with predictions. Rather than class probabilities, a model designed for EDL outputs parameters for a distribution over these probabilities, referred to as a second-order distribution. Below, we first summarize this framework for a nominal problem of multi-label classification. We then extend this framework for 3D object detection.

### 3.1 PRELIMINARIES OF EVIDENTIAL DEEP LEARNING (EDL)

**Model architecture.** Consider a $C$-class multi-label classification problem where the neural network would generate a $C$-dimensional probability vector $\mathbf{e} \in \mathbb{R}^C$. To adapt a neural network for EDL, we modify the head to generate two $C$-dimensional vectors $\mathbf{e}^a, \mathbf{e}^b \in \mathbb{R}^C$. The first vector represents positive "evidence" for each class, $\boldsymbol{\alpha}_j := \mathrm{softplus}(\mathbf{e}^a_j) + 1$ (i.e., indicating that "the model input belongs to the $j$th class"), and the second vector represents negative evidence, $\boldsymbol{\beta}_i := \mathrm{softplus}(\mathbf{e}^b_j) + 1$ (i.e., "the model input does not belong to the $j$th class"). Here, $\boldsymbol{\alpha}_j$ and $\boldsymbol{\beta}_j$ represent the parameters of $\mathrm{Beta}(\boldsymbol{\alpha}_j, \boldsymbol{\beta}_j)$ distributions generating the probabilities for the $i$th class; further, $\mathrm{softplus}(x) := \ln(1 + e^x)$ ensures $\boldsymbol{\alpha}, \boldsymbol{\beta} > \mathbf{0}$. Consequently, the predicted probability for the $i$th class is $P(y_j = 1 | \mathbf{x}) := \frac{\boldsymbol{\alpha}_j}{\boldsymbol{\alpha}_j + \boldsymbol{\beta}_j}$, and the model's uncertainty is $U(\mathbf{x}) := \frac{1}{\boldsymbol{\alpha}_j + \boldsymbol{\beta}_j}$ (Jsang, 2018).

**Loss function.** To train a model for multi-label classification using EDL, the loss function is derived by computing the Bayes risk with respect to the class predictor. Given the $i$th data point, the probability of class $j$ is modeled with a Beta distribution $\mathrm{Beta}(\boldsymbol{\alpha}_{ij}, \boldsymbol{\beta}_{ij})$. The loss is:

Figure 2: **Model architecture with EDL Heatmap Head.** We replace the standard heatmap head with an Evidential Deep Learning (EDL) head, which predicts both object presence probabilities and uncertainty by outputting $\boldsymbol{\alpha}_i$ and $\boldsymbol{\beta}_i$ for each BEV cell.

$$\mathcal{L}_i(\Theta) := \int \left[ \sum_{j=1}^{C} -y_{ij} \log(p_{ij}) \right] \frac{1}{B(\boldsymbol{\alpha}_{ij}, \boldsymbol{\beta}_{ij})} \prod_{j=1}^{C} p_{ij}^{\boldsymbol{\alpha}_{ij}-1} (1-p_{ij})^{\boldsymbol{\beta}_{ij}-1} \, \mathrm{d}\mathbf{p}_i \tag{1}$$

$$= \sum_{j=1}^{C} \left[ y_{ij} \left( \psi(\boldsymbol{\alpha}_{ij} + \boldsymbol{\beta}_{ij}) - \psi(\boldsymbol{\alpha}_{ij}) \right) + (1-y_{ij}) \left( \psi(\boldsymbol{\alpha}_{ij} + \boldsymbol{\beta}_{ij}) - \psi(\boldsymbol{\beta}_{ij}) \right) \right], \tag{2}$$

where $B(\boldsymbol{\alpha}_{ij}, \boldsymbol{\beta}_{ij}) = \frac{\Gamma(\boldsymbol{\alpha}_{ij})\Gamma(\boldsymbol{\beta}_{ij})}{\Gamma(\boldsymbol{\alpha}_{ij}+\boldsymbol{\beta}_{ij})}$ is the Beta function, and $\psi(\cdot)$ is the *digamma* function (the logarithmic derivative of the gamma function, i.e., $\psi(x) := \frac{\mathrm{d}}{\mathrm{d}x} \ln \Gamma(x) = \frac{\Gamma'(x)}{\Gamma(x)}$. We provide the derivation of equation 2 in Appendix A.2.

### 3.2 EVIDENTIAL LEARNING FOR 3D DETECTION

While equation 2 is effective for simple tasks such as image classification (Sensoy et al., 2018) and image-based action recognition (Bao et al., 2021; Zhao et al., 2023), it is not immediately effective for detection tasks. Specifically, in 3D object detection, there are a variable number of detections consisting of a class prediction and object bounding box coordinates. This necessitates an uncertainty estimation method that can capture both class uncertainty (i.e., what type of object is detected) and location uncertainty (i.e., where the object is located). Moreover, the inherent imbalance towards the negative class can heavily bias uncertainty estimators, as most detections correspond to background, leading to underconfidence in positive detections.

To overcome these challenges, we propose to capture uncertainty at the Bird's Eye View (BEV) level (Zhou et al., 2019; Ma et al., 2021; Chen et al., 2023), which allows us to simultaneously account for both the object's 3D position and class. Additionally, we adapt a customized EDL loss function that mitigates the negative class imbalance.

**Model architecture.** In 3D object detection, a dedicated heatmap head can predict the probabilities of object centers from a BEV representation of scene (Chen et al., 2023). In our approach, we replace the standard heatmap head with an EDL head, as illustrated in Fig. 2. Specifically, we follow the multiclass EDL model setup discussed in Sec. 3.1. Instead of predicting $C$ probability values for each BEV cell, we now predict $\boldsymbol{\alpha}_i$ and $\boldsymbol{\beta}_i$, enabling the model to estimate not only the probability of an object's presence but also the uncertainty associated with the prediction. Technically, we double the number of output dimensions, treating the first $C$ as $\boldsymbol{\alpha}_i$ and the second $C$ as $\boldsymbol{\beta}_i$.

**Loss function for 3D.** Common loss functions for EDL, as discussed in Section 3.1, are not directly applicable to object detection due to high class imbalance. In 3D object detection, especially for center-based methods, specialized techniques like Gaussian Focal Loss (GFL) are often employed to address these imbalances (Law & Deng, 2018; Zhou et al., 2019). GFL accounts for areas near object centers using a Gaussian-distributed ground truth heatmap, enhancing localization precision. For further details on GFL and related methods, we refer the reader to Appendix A.1.

To adapt EDL for 3D object detection, we propose a combined loss function: $\mathcal{L} = \sum_{i=1}^{S} (\mathcal{L}_i^{\text{EDL}} + \lambda \mathcal{L}_i^{\text{Reg}})$, where $S$ is the number of training scenes, and $\lambda \geq 0$ is a regularization parameter. Below, we discuss each component of this loss in detail. We begin by introducing the first term, $\mathcal{L}_i^{\text{EDL}}$,

(a) Scene-level Uncertainty          (b) Box-level Uncertainty

Figure 3: **Uncertainty at different levels. (a)** Scene-level uncertainty aggregates uncertainty values across all BEV cells in a scene to produce an overall uncertainty score, which help detect OOD scenes. **(b)** Box-level uncertainty focuses on each predicted bounding box's uncertainty using ROI pooling, allowing for the identification of poorly localized bounding boxes.

which is defined as follows:

$$\mathcal{L}_i^{\text{EDL}} := \sum_{j=1}^{C} \Big[ y_{ij} \left( \psi(\boldsymbol{\alpha}_{ij} + \boldsymbol{\beta}_{ij}) - \psi(\boldsymbol{\alpha}_{ij}) \right) \cdot \left( 1 - \boldsymbol{\alpha}_{ij}/(\boldsymbol{\alpha}_{ij} + \boldsymbol{\beta}_{ij}) \right)^{\gamma}$$

$$+ (1 - y_{ij}) \left( \psi(\boldsymbol{\alpha}_{ij} + \boldsymbol{\beta}_{ij}) - \psi(\boldsymbol{\beta}_{ij}) \right) \cdot \left( \boldsymbol{\alpha}_{ij}/(\boldsymbol{\alpha}_{ij} + \boldsymbol{\beta}_{ij}) \right)^{\gamma} \cdot (1 - \hat{y}_i)^{\eta} \Big]. \quad (3)$$

The above loss function is composed of two main terms:

• The first term in the sum corresponds to the BEV grid cells where an actual object center is located (i.e., $y_{ij} = 1$). Since we are training an EDL model, we compute the digamma-based Bayes risk loss for each of these cells. This risk is further scaled by a GFL-based factor $(1 - \boldsymbol{\alpha}_{ij}/(\boldsymbol{\alpha}_{ij} + \boldsymbol{\beta}_{ij}))^{\gamma}$, which helps reduce the impact of well-classified examples and focus on harder, misclassified examples during training. This is particularly important in object detection, where a large portion of the training examples can be relatively easy (e.g., background regions), and the model needs to pay more attention to difficult examples, such as object boundaries.

• The second term handles locations where there are no objects (i.e., $y_{ij} = 0$). Here, the Bayes risk is computed similarly, also weighted by a GFL-based factor focusing more on difficult negative examples. In addition, we apply a discounting term $(1 - \hat{y}_i)^{\eta}$, which reduces the penalty for predictions made in the vicinity of an object's center. This means that if the model predicts a high probability for a cell being an object's center when it is not, but the cell is close to the object center, the penalty is smaller. In contrast, if the model predicts high probabilities for locations far away from any objects, the penalty is much larger. This encourages the model to be more precise in localizing object centers while tolerating small errors near the true center.

**Regularization term.** In our method, we include a regularization term to manage uncertainty by penalizing the model when it generates incorrect or overconfident predictions, following a similar strategy to the one used in the original EDL framework (Sensoy et al., 2018). The goal is to minimize misleading evidence, particularly when the model makes incorrect predictions. In Sensoy et al. (2018); Amini et al. (2020), regularization is applied by encouraging the model to revert to a uniform Dirichlet prior (representing high uncertainty) when predictions are incorrect, thereby penalizing misleading evidence and avoiding overconfident mistakes. In our approach, we use adjust evidences $\tilde{\boldsymbol{\alpha}}_i$ and $\tilde{\boldsymbol{\beta}}_i$ for such regularization as follows:

$$\tilde{\boldsymbol{\alpha}}_i := \mathbf{y}_i + (1 - \mathbf{y}_i) \odot \boldsymbol{\alpha}_i, \quad \tilde{\boldsymbol{\beta}}_i := (1 - \mathbf{y}_i) + \mathbf{y}_i \odot \boldsymbol{\beta}_i, \quad \text{where } \odot \text{ is the Hadamard product.} \quad (4)$$

These adjustments adapt based on the correctness of the predictions, aiming to minimize evidence for incorrect predictions. To achieve this, we introduce a divergence-based regularization term that minimizes the Kullback-Leibler divergence between the modified Beta distribution $\text{Beta}(\tilde{\boldsymbol{\alpha}}_i, \tilde{\boldsymbol{\beta}}_i)$ and the prior $\text{Beta}(\mathbf{1}, \mathbf{1})$, which encourages the model to express total uncertainty (i.e., a uniform distribution) when necessary. In other words, we have $\mathcal{L}_i^{\text{Reg}} := \sum_{j=1}^{C} \text{KL} \left( \text{Beta}(\tilde{\boldsymbol{\alpha}}_j, \tilde{\boldsymbol{\beta}}_j) \,\|\, \text{Beta}(\mathbf{1}, \mathbf{1}) \right)$.

Our regularization term can be rewritten as follows:

$$\mathcal{L}_i^{\text{Reg}} = \sum_{j=1}^{C} \Big[ (\tilde{\boldsymbol{\alpha}}_{ij}-1)(\psi(\tilde{\boldsymbol{\alpha}}_{ij})-\psi(\tilde{\boldsymbol{\alpha}}_{ij}+\tilde{\boldsymbol{\beta}}_{ij})) + (\tilde{\boldsymbol{\beta}}_{ij}-1)(\psi(\tilde{\boldsymbol{\beta}}_{ij})-\psi(\tilde{\boldsymbol{\alpha}}_{ij}+\tilde{\boldsymbol{\beta}}_{ij})) - \log\Big(B(\tilde{\boldsymbol{\alpha}}_{ij},\tilde{\boldsymbol{\beta}}_{ij})\Big) \Big]$$

(5)

Using equation 5 as regularization term prevents the model from being overconfident in its predictions. We provide the derivation of equation 5 in Appendix A.3.

## 4 EXPERIMENTS

We demonstrate the value of EDL-based uncertainty estimates across three different downstream applications: (i) detecting out-of-distribution scenes; (ii) estimating the localization quality of predicted bounding boxes; and (iii) identifying missing detections in a scene. Our estimated uncertainties consistently yield better downstream performance than existing baselines. Finally, we integrate these tasks into an auto-labeling pipeline where estimated pseudo-labels from a 3D object detector are flagged and verified accordingly to uncertainty scores. These "corrected" pseudo-labels are used to train a downstream detector that achieves improved performance over baselines that use unverified pseudo-labels for training.

### 4.1 EXPERIMENTAL SETUP

**Dataset and metric.** We evaluate our approach on the *nuScenes* and *Waymo* 3D detection datasets.

The *nuScenes Dataset* (Caesar et al., 2020) is a comprehensive large-scale driving dataset with 1,000 scenes of multi-modal data, including 32-beam LiDAR at 20 FPS and images from six different camera views. We explore two settings: *LiDAR-only* and *LiDAR-Camera fusion*. We evaluate detection performance on mean average precision (mAP) and the nuScenes detection score (NDS), defined by averaging the matching thresholds of center distance $\mathbb{D} = \{0.5, 1., 2., 4.\}$ meters.

The *Waymo Open Dataset* (Sun et al., 2020) includes 798 scenes for training and 202 scenes for validation. The nuScenes and Waymo Open datasets were recorded in different locations over different vehicles. They differ significantly in terms of detection range, scene composition, and sensor configuration, making the Waymo dataset a suitable choice for out-of-distribution (OOD) testing.

**3D Detection Baselines.** We use two 3D detection models for the *nuScenes* dataset: 1) *Focal-Former3D* (Chen et al., 2023), a state-of-the-art architecture for both Lidar and Lidar + Camera configurations, and 2) *DeformFormer3D*, a Lidar-based architecture built upon DETR (Carion et al., 2020) and Deformable DETR (Zhu et al., 2021). In our experiments, we refer to FocalFormer3D Lidar experiments as *FF (L)*, FocalFormer3D Lidar+Camera experiments as *FF (L+C)*, and DeformFormer3D Lidar experiments as *DF (L)*.

**Implementation details.** Our implementation is built on the open-source MMDetection3D codebase (Contributors, 2020). For the LiDAR backbone, we use CenterPoint-Voxel as the feature extractor for point clouds. The multi-stage heatmap encoder operates in 3 stages, producing 600 queries by default. Data augmentation techniques include random double flips along the $X$ and $Y$ axes, random global rotation within the range of $[-\pi/4, \pi/4]$, scaling randomly between $[0.9, 1.1]$, and random translations with a standard deviation of $0.5$ across all axes. Training is conducted with a batch size of 16 across eight V100 GPUs. Similar to Amini et al. (2020), we set the regularization parameter $\lambda$ in the loss function to $10^{-4}$ to prevent over-regularization of the model.

**Uncertainty Baselines.** We compare against the standard *Entropy* (Malinin & Gales, 2018) baseline, along with recent sample-based approaches: MC-Dropout (Gal & Ghahramani, 2016) (*MC-D*), Deep Ensembles (Lakshminarayanan et al., 2017) (*DeepE*), BatchEnsemble (Wen et al., 2020) (*BatchE*), Masksembles (Durasov et al., 2021) (*MaskE*), and Packed-Ensembles (Laurent et al., 2023) (*PackE*), which are known for producing state-of-the-art uncertainty estimates (Ashukha et al., 2020; Postels et al., 2022). For all four sampling-based methods, we use five samples to estimate uncertainty at inference time, a setting shown to work well across multiple tasks (Durasov et al., 2021; Wen et al., 2020; Malinin & Gales, 2018).

Table 1: **Scene OOD detection ROC- and PR-AUCs evaluation.** The best result in each category is in **bold** and the second best is in **bold**. *Ours* outperforms the second-best on average by 0.09 *ROC-AUC* & 0.16 *PR-AUC*, respectively.

|  | *Entropy* | *MC-DP* | *BatchE* | *MaskE* | *PackE* | *DeepE* | *Ours* | Model |
|---|---|---|---|---|---|---|---|---|
| *ROC-AUC* | 0.4893 | 0.4875 | 0.4972 | 0.4872 | 0.5360 | 0.5059 | **0.6282** | FF (L) |
| *PR-AUC* | 0.4815 | 0.4917 | 0.4887 | 0.4836 | 0.5292 | 0.4982 | **0.7168** | |
| *ROC-AUC* | 0.4074 | 0.5167 | 0.6020 | 0.5910 | 0.4214 | 0.5374 | **0.7694** | FF (L+C) |
| *PR-AUC* | 0.4316 | 0.5310 | 0.5594 | 0.5480 | 0.5314 | 0.4325 | **0.7424** | |
| *ROC-AUC* | 0.4378 | 0.5134 | 0.4485 | 0.5447 | 0.3494 | **0.6622** | 0.5806 | DF (L) |
| *PR-AUC* | 0.4589 | 0.5100 | 0.4616 | 0.5177 | 0.4016 | **0.5894** | 0.5495 | |
| *Average ROC-AUC* | 0.4448 | 0.5059 | 0.5159 | 0.5410 | 0.4356 | 0.5685 | **0.6594** | N/A |
| *Average PR-AUC* | 0.4573 | 0.5109 | 0.5032 | 0.5164 | 0.4874 | 0.5067 | **0.6696** | |

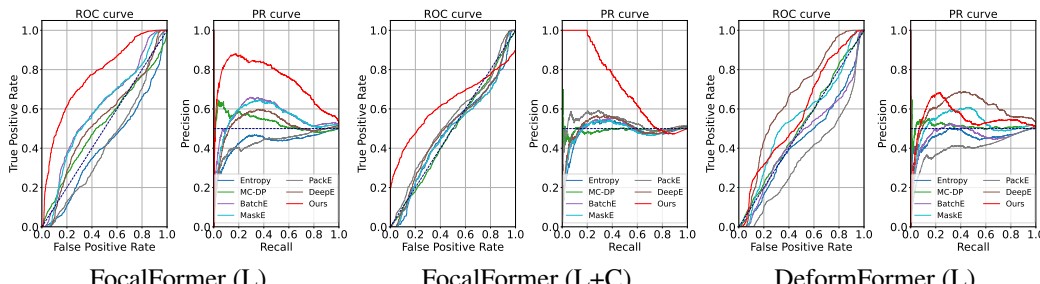

| FocalFormer (L) | FocalFormer (L+C) | DeformFormer (L) |

Figure 4: **Scene out-of-distribution detection ROC and PR curves evaluation.** ROC and PR curves for the OOD detection task using the uncertainty measure described in Section 4.2. A higher position of the curve indicates a better ability of the uncertainty measure to detect OOD scenes. Our uncertainty measure outperforms other methods by a significant margin across various setups.

## 4.2 DETECTING OOD SCENES

To demonstrate the quality of uncertainty estimates generated by our framework, we apply it to detect scene-level OOD samples, under the assumption that a network will be more uncertain about predictions made on OOD scenes than on scenes in the training distribution. As in Malinin & Gales (2018); Durasov et al. (2021), given both in-distribution and OOD samples, we classify high-uncertainty samples as OOD and rely on standard classification metrics (i.e., ROC and PR AUC) to quantify the classification performance. To estimate the uncertainty score for the whole scenes, we first produce uncertainty values for each class and BEV cell, then we average the generated uncertainty values across all cells in a scene, as it is illustrated in Fig. 3 (a).

We train the model on the training set of the nuScenes dataset and consider scenes from the nuScenes test set as in-distribution samples. At the same time, we take scenes from the Waymo test set and treat them as OOD samples since the nuScenes and Waymo datasets have different statistics, objects, and lidar specifications. This leads to a significant drop in performance (Wang et al., 2020) when a model trained on one dataset is applied to another. We report results in terms of ROC and PR curves in Fig. 4, and we report aggregated ROC and PR-AUC in Tab 1.

Our approach significantly outperforms other methods in detecting OOD scenes when using Focal-Former (L) and (L+C), as shown in Fig. 4 and Tab. 1. This is due to FocalFormer's multi-stage procedure for BEV embedding generation, which produces richer and more detailed feature representations. In contrast, DeformFormer uses a single-stage process, limiting its ability to fully refine spatial and object-level features, which affects the performance of most uncertainty baselines. This could explain its slightly lower performance, as its simpler approach leads to less detailed BEV embeddings.

Table 2: **Detection of erroneous boxes ROC- and PR-AUCs evaluation.** The best result in each category is in **bold** and the second best is in **bold**. *Ours* outperforms the second-best on average by 0.06 *ROC-AUC* & 0.02 *PR-AUC*, respectively.

|  | *Entropy* | *MC-DP* | *BatchE* | *MaskE* | *PackE* | *DeepE* | *Ours* | Model |
|---|---|---|---|---|---|---|---|---|
| *ROC-AUC* | 0.5615 | 0.5515 | 0.5829 | 0.5777 | 0.5890 | 0.5923 | **0.6329** | |
| *PR-AUC* | 0.3224 | 0.2874 | 0.3326 | 0.3253 | 0.3259 | 0.3340 | **0.3646** | FF (L) |
| *Corr* | 0.1623 | 0.2227 | 0.2151 | 0.2270 | 0.2604 | 0.2304 | **0.3142** | |
| *ROC-AUC* | 0.5609 | 0.5692 | 0.5673 | 0.5663 | 0.5642 | 0.5478 | **0.6148** | |
| *PR-AUC* | 0.3409 | 0.3328 | 0.3334 | 0.2993 | 0.3513 | 0.3219 | **0.3529** | FF (L+C) |
| *Corr* | 0.1379 | 0.1909 | 0.1777 | 0.1614 | 0.1800 | 0.1270 | **0.3119** | |
| *ROC-AUC* | 0.5319 | 0.5512 | 0.5760 | 0.5580 | 0.5565 | 0.5656 | **0.6229** | |
| *PR-AUC* | 0.3227 | 0.3286 | 0.3503 | 0.3464 | 0.3373 | 0.3518 | **0.3726** | DF (L) |
| *Corr* | 0.1139 | 0.1401 | 0.1584 | 0.1495 | 0.1591 | 0.1766 | **0.2716** | |
| *Average ROC-AUC* | 0.5514 | 0.5573 | 0.5754 | 0.5673 | 0.5699 | 0.5686 | **0.6235** | |
| *Average PR-AUC* | 0.3287 | 0.3163 | 0.3388 | 0.3237 | 0.3382 | 0.3359 | **0.3634** | N/A |
| *Average Corr* | 0.1380 | 0.1846 | 0.1837 | 0.1793 | 0.1998 | 0.1780 | **0.2992** | |

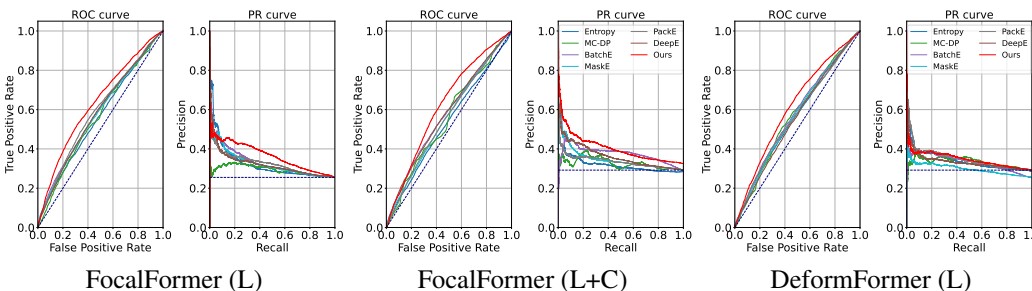

FocalFormer (L)          FocalFormer (L+C)          DeformFormer (L)

Figure 5: **ROC and PR curve evaluation for the detection of erroneous boxes.** ROC and PR curves for the erroneous box detection task using the uncertainty measure in Section 4.3. A higher position of the curve indicates a better ability of the uncertainty measure to detect erroneous boxes predicted by the model. Our uncertainty measure outperforms baselines across various setups.

### 4.3 IDENTIFYING BOUNDING-BOX LOCALIZATION ERRORS

Uncertainty estimates produced via our framework can be used to predict whether the model will generate poorly localized bounding boxes. To classify a poorly localized bounding box, we calculate the Intersection-over-Union (IoU) between predicted and ground truth boxes; we denote a poorly localized "erroneous" box if the IoU is below an arbitrary threshold $\tau \in [0, 1]$, and "accurate" otherwise. We set $\tau = 0.3$, which ensures that boxes classified as *erroneous* exhibit sufficiently low overlap with the ground truth, indicating poor localization accuracy. We then transform the task of detecting erroneous boxes into a binary classification based on the overall uncertainty in the predicted box, allowing us to compute standard ROC and PR curves (and their corresponding AUCs).

To assign an uncertainty value to each predicted box, we follow a consistent procedure across all uncertainty methods. After generating an uncertainty map for each BEV cell and class, i.e., $\hat{u} \in \mathbb{R}^{C \times H \times D}$ where $C$ is the number of classes, and $H$ and $D$ are the BEV height and depth, respectively. For any predicted bounding box, we then collect the set of uncertainty values within the given box $\{\hat{u}_b^i\}$ (see Fig. 3 (b)), where $i$ corresponds to the class index and $b$ corresponds to the cell. Finally, we aggregate these estimates into a single uncertainty score for the predicted bounding box $u_b = \min_i \hat{u}_b^i$.

We use the model trained on the nuScenes training set and run inference on the test set. After generating predicted boxes for all scenes in the test set, we compute the uncertainty and IoU for each box, followed by the AUC metrics as described earlier. The results are reported in Fig. 5 and Tab. 2. Our approach consistently outperforms other uncertainty baselines by $5 - 10\%$.

## 4.4 IDENTIFYING UNDETECTED OBJECTS

We also focus on recall, which can be reframed as minimizing the number of false negative detections, and is an important metric for a 3D detector suitable for deployment in autonomous vehicles. As discussed in Sec. 3.2, modern 3D object detection models typically include a stage where the model estimates the probability of an object being in a specific BEV cell. If a cell has a low predicted probability, it is less likely that any bounding box will be generated with its center in that BEV cell. This means that if the cell actually contains a ground truth object, it will not be detected, resulting in a false negative. In some cases, our heatmap may assign low probabilities to locations where there is an actual object, but this is often accompanied by higher uncertainty in the prediction. This indicates the model's uncertainty in identifying objects in certain areas. As a result, we are motivated to leverage these predicted uncertainties to identify potentially missed objects and improve detection performance in such challenging scenarios.

To address this, we implemented a separate head $\mathcal{M}^{\text{miss}}$ that processes the BEV embeddings $\mathbf{e}_i$, predicted probabilities $\mathbf{p}_i$, and uncertainty $\mathbf{u}_i$ from our EDL head for each BEV cell, using the concatenated vector $[\mathbf{e}_i, \mathbf{p}_i, \mathbf{u}_i]$ to estimate the confidence of potentially missed objects in the given cells as $\mathbf{p}_i^{\text{miss}} = \mathcal{M}^{\text{miss}}([\mathbf{e}_i, \mathbf{p}_i, \mathbf{u}_i])$. Technically, we only consider BEV cells where the heatmapping head produces low probabilities ($\mathbf{p}_i$ less than 5% in our experiments) as candidates for locations where objects could have been missed. This threshold was chosen because cells with such low probabilities are excluded from the second stage, and no bounding boxes are generated for them in the final predictions. By focusing on these ignored cells, we aim to identify potential false negatives that would otherwise go undetected by the model. The $\mathcal{M}^{\text{miss}}$ head is trained using the same targets, loss, and training procedure as the original heatmap head, with the only difference being that it is trained on cells with low probability $\mathbf{p}_i$ and uses $\mathbf{p}_i$ and $\mathbf{u}_i$ in addition to $\mathbf{e}_i$ as input.

As in Sec. 4.3, we use the model trained on the nuScenes training set and run inference on the test set. To evaluate the quality of the newly detected locations with potential missed objects, we followed this evaluation procedure: First, we generated the final bounding box predictions and compared them against the ground truth boxes from the test set to identify the missed ground truth boxes. Next, using our new probability head, we predicted $\mathbf{p}_i^{\text{miss}}$ for each BEV cell and identified 15 new locations with potential missed objects (this number was chosen to balance precision and recall), selecting the locations with the highest $\mathbf{p}_i^{\text{miss}}$ scores. If a missed ground truth box was found within a radius of $d$ meters ($d = 2$ and $d = 4$ in our experiments) from the predicted location, it was considered a true positive detection. By treating this as a classification task, we calculated precision, recall, and F1-score for these detections, and the results are reported in Tab. 3. As in the previous tasks, our method significantly outperformed other approaches.

## 4.5 PUTTING IT ALL TOGETHER: AUTO-LABELING WITH VERIFICATION

Finally, we introduce an auto-labeling framework with verification that leverages uncertainty estimates to improve 3D object detection. Auto-labeling is crucial in contexts where acquiring annotated data is expensive or time-consuming, allowing models to generate labels for unlabeled data (Elezi et al., 2022; Beck et al., 2024). However, traditional auto-labeling approaches lack mechanisms to verify the reliability of the generated labels, leading to potential noise or mislabeling.

Our framework addresses this issue by incorporating uncertainty-driven verification as a core component. After generating initial pseudo-targets (bounding boxes) with a 3D detector, we estimate uncertainties at multiple levels: (1) scene-level (Sec. 4.2) — identifying scenes with high uncertainty and relabeling the entire scene, (2) box-level (Sec. 4.3) — detecting pseudo-labels with high uncertainty for relabeling or verification, and (3) missed objects (Sec. 4.3) — identifying potential missed objects through uncertainty and labeling them accordingly. To ensure balanced coverage, we allocate an equal budget across these categories, resulting in approximately 10,000 boxes and 30,000 labels in total.

We compare our approach to two baselines in two configurations. For the first baseline (referred to as "$N$k-R" in Tab. 4), we use $N$ thousand scenes (where $N \in \{10, 20\}$) from the nuScenes training set, train on them, and then evaluate on the test set. The second baseline (referred to as "$N$k-P") also trains on $N$ thousand scenes but generates pseudo-labels for the unlabeled portion of the training set, retrains on the entire dataset, and then evaluates on the test set. Our method "$N$k-U", on the other hand, begins by training on $N - 1$ thousand scenes from the training set, applies our uncertainty-

Table 3: **Missed object detection evaluation.** The best result in each category is in **bold** and the second best is in **bold**. *Ours* outperforms others by a significant margin.

| | *Entropy* | *MC-DP* | *BatchE* | *MaskE* | *PackE* | *DeepE* | *Ours* | d, m | M |
|---|---|---|---|---|---|---|---|---|---|
| *Precision* | 0.0766 | 0.0408 | 0.1354 | 0.1015 | 0.0459 | 0.0792 | **0.1512** | | |
| *Recall* | 0.0367 | 0.0191 | 0.0614 | 0.0491 | 0.0219 | 0.0379 | **0.0735** | 2 | |
| *F1-Score* | 0.0496 | 0.0260 | 0.0758 | 0.0662 | 0.0296 | 0.0513 | **0.0989** | | FF (L) |
| *Precision* | 0.1367 | 0.0641 | 0.1675 | 0.1662 | 0.0924 | 0.1410 | **0.2352** | | |
| *Recall* | 0.0667 | 0.0191 | 0.0833 | 0.0829 | 0.0444 | 0.0688 | **0.1135** | 4 | |
| *F1-Score* | 0.0897 | 0.0260 | 0.1113 | 0.1106 | 0.0600 | 0.0924 | **0.1531** | | |
| *Precision* | 0.0476 | 0.0341 | 0.0363 | 0.0786 | 0.0350 | 0.0478 | **0.1032** | | |
| *Recall* | 0.0207 | 0.0149 | 0.0173 | 0.0375 | 0.0152 | 0.0208 | **0.0463** | 2 | |
| *F1-Score* | 0.0288 | 0.0208 | 0.0234 | 0.0508 | 0.0212 | 0.0290 | **0.0639** | | FF (L+C) |
| *Precision* | 0.0970 | 0.0848 | 0.0813 | 0.1418 | 0.0821 | 0.0972 | **0.1501** | | |
| *Recall* | 0.0428 | 0.0371 | 0.0392 | 0.0688 | 0.0360 | 0.0428 | **0.0695** | 4 | |
| *F1-Score* | 0.0593 | 0.0516 | 0.0529 | 0.0926 | 0.0501 | 0.0595 | **0.0950** | | |
| *Precision* | 0.0430 | 0.0182 | 0.0671 | 0.0562 | 0.0459 | 0.0387 | **0.1167** | | |
| *Recall* | 0.0200 | 0.0084 | 0.0316 | 0.0262 | 0.0219 | 0.0180 | **0.0557** | 2 | |
| *F1-Score* | 0.0273 | 0.0115 | 0.0429 | 0.0357 | 0.0296 | 0.0246 | **0.0754** | | DF (L) |
| *Precision* | 0.0810 | 0.0788 | 0.1101 | 0.0999 | 0.0924 | 0.0810 | **0.1743** | | |
| *Recall* | 0.0382 | 0.0364 | 0.0525 | 0.0467 | 0.0444 | 0.0382 | **0.0855** | 4 | |
| *F1-Score* | 0.0519 | 0.0498 | 0.0711 | 0.0637 | 0.0600 | 0.0519 | **0.1147** | | |
| *Average Precision* | 0.0803 | 0.0534 | 0.0996 | 0.1073 | 0.0656 | 0.0808 | **0.15151** | | |
| *Average Recall* | 0.0375 | 0.0225 | 0.0475 | 0.0518 | 0.0306 | 0.0377 | **0.0740** | N/A | N/A |
| *Average F1-Score* | 0.0511 | 0.0309 | 0.0629 | 0.0699 | 0.0418 | 0.0514 | **0.1002** | | |

Table 4: **NuScenes auto-labeling results.** We compare our uncertainty-based verification method against two baselines: standard training on the smaller training set and auto-labeling without uncertainty-based verification. *FT* represents training on the entire dataset, which we consider as an upper bound for quality. The results show that our approach consistently outperforms both baselines, achieving higher mAP and NDS scores across all configurations, with significant relative improvements over the auto-labeling without uncertainty baseline, as shown in the "Imp, %" column.

| | *10k R* | *10k P* | *10k U* | *20k R* | *20k P* | *20k U* | *FT* | Imp, % | Model |
|---|---|---|---|---|---|---|---|---|---|
| mAP ($\uparrow$) | 0.5881 | 0.5942 | 0.5964 | 0.6299 | 0.6321 | 0.6398 | 0.6636 | +0.785 | FF (L) |
| NDS ($\uparrow$) | 0.5474 | 0.5539 | 0.5577 | 0.5827 | 0.5859 | 0.6057 | 0.7093 | +1.829 | |
| mAP ($\uparrow$) | 0.6664 | 0.6767 | 0.6797 | 0.6966 | 0.6975 | 0.7002 | 0.7049 | +0.412 | FF (L+C) |
| NDS ($\uparrow$) | 0.5868 | 0.5902 | 0.5931 | 0.6063 | 0.6057 | 0.6077 | 0.7314 | +0.410 | |
| mAP ($\uparrow$) | 0.5847 | 0.6091 | 0.6115 | 0.6429 | 0.6100 | 0.6151 | 0.6553 | +0.618 | DF (L) |
| NDS ($\uparrow$) | 0.5426 | 0.5734 | 0.5769 | 0.5777 | 0.5734 | 0.5782 | 0.7071 | +0.722 | |

based verification to relabel 1,000 scenes (or 30,000 boxes) from the unlabeled set, retrains on the fully auto-labeled dataset, and evaluates on the test set. As shown in Tab. 4, our method significantly improves regular auto-labeling by incorporating uncertainty into the label verification pipeline.

## 5 CONCLUSION

We have introduced an Evidential Deep Learning framework for uncertainty estimation in 3D object detection using Bird's Eye View representations. Our method improves out-of-distribution detection, bounding box quality, and missed object identification, all while being computationally efficient. By incorporating uncertainty into the auto-labeling pipeline, we achieve significant gains in mAP and NDS on the nuScenes dataset, which also leads to enhanced auto-label quality.

One limitation of our current approach is that we only update the head of the model. Expanding to end-to-end learning could potentially yield better results by enabling the model to learn uncertainty more holistically across all layers. Despite this, our method remains computationally efficient, offering a practical alternative to deep ensembles, and is well-suited for large-scale, real-time applications like autonomous driving. The ability to integrate seamlessly into an auto-labeling pipeline highlights its practical utility, reducing the cost and effort of manual labeling while improving data quality in scenarios where high-quality annotations are critical.

ETHICS STATEMENT

The 3D detection models discussed in this work are core components towards applications in autonomous driving and robotics which interface with the human world. Accurately quantifying model uncertainty is essential, as uncertainty estimates can be used to determine if the model predictions are unreliable and the downstream technologies warrant human intervention. Our work specifically targets this problem. Furthermore, this research is reliant on both the quality and quantity of training data to mitigate biased estimates or data privacy concerns.

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

# A APPENDIX

## A.1 OBJECT DETECTION LOSSES

**Focal Loss.** Focal Loss (Lin et al., 2017) is designed to address class imbalance by down-weighting well-classified examples and focusing on harder, misclassified examples. The Focal Loss function is defined as:

$$\mathcal{L}_{\text{FL}}(p) = -(1 - p)^\gamma \log(p),$$

where $p$ is the model's estimated probability for the correct class, and $\gamma$ is a tunable parameter that controls the focus on harder examples. When $\gamma = 0$, Focal Loss reduces to standard cross-entropy loss. As $\gamma$ increases, the loss function places greater emphasis on difficult examples, which is useful in contexts like object detection where class imbalance is common. This is particularly effective when most examples are easy negatives (e.g., background) and would otherwise dominate the loss.

**Gaussian Focal Loss.** Gaussian Focal Loss (GFL) (Law & Deng, 2018; Zhou et al., 2019) is a variant of Focal Loss specifically designed for 3D object detection. In this approach, object centers are represented using a Gaussian heatmap, and the loss function focuses on the points near the center of the object. The Gaussian Focal Loss function is given by:

$$\mathcal{L}_{\text{GFL}} = -(1 - \hat{y})^\eta \log(p),$$

where $\hat{y}$ is the Gaussian-distributed ground truth centered around the object, and $p$ is the predicted probability. The term $(1 - \hat{y})^\eta$ down-weights points far from the object's center, ensuring that the model focuses on improving localization precision for points near the center. The parameter $\eta$ controls the degree to which points further from the center are down-weighted.

## A.2 DERIVATION OF THE LOSS

**Loss function.** We aim to derive the loss function for multi-label classification using the Beta distribution by computing the Bayes risk with respect to the class predictor. The probability of class $j$ for instance $i$ is modeled as a Beta distribution, $\text{Beta}(\boldsymbol{\alpha}_{ij}, \boldsymbol{\beta}_{ij})$. The resulting loss is given by:

$$\mathcal{L}_i(\Theta) := \int \left[ \sum_{j=1}^{C} -y_{ij} \log(p_{ij}) \right] \frac{1}{B(\boldsymbol{\alpha}_{ij}, \boldsymbol{\beta}_{ij})} \prod_{j=1}^{C} p_{ij}^{\boldsymbol{\alpha}_{ij}-1} (1 - p_{ij})^{\boldsymbol{\beta}_{ij}-1} \, d\mathbf{p}_i, \qquad (6)$$

where $B(\boldsymbol{\alpha}_{ij}, \boldsymbol{\beta}_{ij})$ is the Beta function, defined as:

$$B(\boldsymbol{\alpha}_{ij}, \boldsymbol{\beta}_{ij}) = \frac{\Gamma(\boldsymbol{\alpha}_{ij}) \Gamma(\boldsymbol{\beta}_{ij})}{\Gamma(\boldsymbol{\alpha}_{ij} + \boldsymbol{\beta}_{ij})}, \qquad (7)$$

and $\Gamma(\cdot)$ is the Gamma function. For each class $j$, we consider the individual term:

$$\mathcal{L}_i(\Theta) = \sum_{j=1}^{C} \int -y_{ij} \log(p_{ij}) \frac{p_{ij}^{\alpha_{ij}-1} (1 - p_{ij})^{\beta_{ij}-1}}{B(\boldsymbol{\alpha}_{ij}, \boldsymbol{\beta}_{ij})} \, dp_{ij}. \qquad (8)$$

The integral term is the expected value of $-\log(p_{ij})$ with respect to a Beta distribution. The expectation of $\log(p_{ij})$ under a Beta distribution $\text{Beta}(\boldsymbol{\alpha}_{ij}, \boldsymbol{\beta}_{ij})$ is given by:

$$\mathbb{E}_{p_{ij} \sim \text{Beta}(\boldsymbol{\alpha}_{ij}, \boldsymbol{\beta}_{ij})} \left[ \log(p_{ij}) \right] = \psi(\boldsymbol{\alpha}_{ij}) - \psi(\boldsymbol{\alpha}_{ij} + \boldsymbol{\beta}_{ij}), \qquad (9)$$

.

Thus, applying the expectation for both $p_{ij}$ and $(1 - p_{ij})$, the loss function becomes:

$$\mathcal{L}_i(\Theta) = \sum_{j=1}^{C} \left[ y_{ij} \left( \psi(\boldsymbol{\alpha}_{ij} + \boldsymbol{\beta}_{ij}) - \psi(\boldsymbol{\alpha}_{ij}) \right) + (1 - y_{ij}) \left( \psi(\boldsymbol{\alpha}_{ij} + \boldsymbol{\beta}_{ij}) - \psi(\boldsymbol{\beta}_{ij}) \right) \right]. \qquad (10)$$

## A.3 DERIVATION OF THE REGULARIZATION TERM

We want to compute the Kullback-Leibler (KL) divergence between the predicted Beta distribution $\text{Beta}(\tilde{\boldsymbol{\alpha}}_i, \tilde{\boldsymbol{\beta}}_i)$ and the prior Beta distribution $\text{Beta}(\mathbf{1}, \mathbf{1})$. The KL divergence between two distributions $P(x)$ and $Q(x)$ is given by:

$$\text{KL}(P\|Q) = \int P(x) \log \frac{P(x)}{Q(x)} \, \mathrm{d}x.$$

The Beta distribution is parameterized by two values, $\alpha$ and $\beta$, and is given by:

$$\text{Beta}(x \mid \alpha, \beta) = \frac{x^{\alpha-1}(1-x)^{\beta-1}}{B(\alpha, \beta)},$$

where $B(\alpha, \beta)$ is the Beta function that normalizes the distribution and is defined as:

$$B(\alpha, \beta) = \frac{\Gamma(\alpha)\Gamma(\beta)}{\Gamma(\alpha+\beta)},$$

with $\Gamma(\cdot)$ being the Gamma function. For two Beta distributions $\text{Beta}(\alpha_1, \beta_1)$ and $\text{Beta}(\alpha_2, \beta_2)$, the KL divergence is computed as:

$$\text{KL}\left(\text{Beta}(\alpha_1, \beta_1)\|\text{Beta}(\alpha_2, \beta_2)\right) = \int_0^1 \text{Beta}(x \mid \alpha_1, \beta_1) \log \frac{\text{Beta}(x \mid \alpha_1, \beta_1)}{\text{Beta}(x \mid \alpha_2, \beta_2)} \, \mathrm{d}x.$$

Substituting the expressions for both Beta distributions, we get:

$$\frac{\text{Beta}(x \mid \alpha_1, \beta_1)}{\text{Beta}(x \mid \alpha_2, \beta_2)} = \frac{x^{\alpha_1-1}(1-x)^{\beta_1-1}B(\alpha_2, \beta_2)}{x^{\alpha_2-1}(1-x)^{\beta_2-1}B(\alpha_1, \beta_1)}.$$

Simplifying, we obtain:

$$\frac{\text{Beta}(x \mid \alpha_1, \beta_1)}{\text{Beta}(x \mid \alpha_2, \beta_2)} = \frac{B(\alpha_2, \beta_2)}{B(\alpha_1, \beta_1)}x^{\alpha_1-\alpha_2}(1-x)^{\beta_1-\beta_2}.$$

Thus, the KL divergence becomes:

$$\begin{aligned}
\text{KL}(\text{Beta}(\alpha_1, \beta_1)\|\text{Beta}(\alpha_2, \beta_2)) = {} & \log \frac{B(\alpha_2, \beta_2)}{B(\alpha_1, \beta_1)} \\
& + (\alpha_1 - \alpha_2) \int_0^1 \text{Beta}(x \mid \alpha_1, \beta_1) \log x \, \mathrm{d}x \\
& + (\beta_1 - \beta_2) \int_0^1 \text{Beta}(x \mid \alpha_1, \beta_1) \log(1-x) \, \mathrm{d}x.
\end{aligned}$$

From known properties of the Beta distribution, we have the following results:

$$\int_0^1 \text{Beta}(x \mid \alpha_1, \beta_1) \log x \, \mathrm{d}x = \psi(\alpha_1) - \psi(\alpha_1 + \beta_1),$$

and

$$\int_0^1 \text{Beta}(x \mid \alpha_1, \beta_1) \log(1-x) \, \mathrm{d}x = \psi(\beta_1) - \psi(\alpha_1 + \beta_1),$$

where $\psi(\cdot)$ is the digamma function, defined as the derivative of the logarithm of the Gamma function, $\psi(x) = \frac{\mathrm{d}}{\mathrm{d}x} \log \Gamma(x)$. Substituting these results into the KL divergence formula, we get the full expression:

$$\text{KL}\left(\text{Beta}(\alpha_1, \beta_1)\|\text{Beta}(\alpha_2, \beta_2)\right) = \log \frac{B(\alpha_2, \beta_2)}{B(\alpha_1, \beta_1)} + (\alpha_1 - \alpha_2)(\psi(\alpha_1) - \psi(\alpha_1 + \beta_1)) + (\beta_1 - \beta_2)(\psi(\beta_1) - \psi(\alpha_1 + \beta_1)).$$

