# OpenReview forum: "Uncertainty Estimation for 3D Object Detection via Evidential Learning"
_ICLR.cc/2025/Conference — Submitted to ICLR 2025_

### Official Review · Reviewer_7H8K · 2024-10-31

**Soundness:** 1
**Presentation:** 2
**Contribution:** 1
**Rating:** 3
**Confidence:** 3

**Summary:**

The paper focuses on uncertainty estimation in 3D object detection. Three tasks are used to testify to the effectiveness of evaluation uncertainty: (1) Out-of-distribution (OOD) scene detection, (2) Bounding-box quality assessment, and (3) Missed objects detection. The uncertainty estimation is conducted by doubling the output channel of existing methods and using the extra channel to estimate the uncertainty value in the BEV. Experimentally, uncertainty estimation is demonstrated for the three tasks mentioned above. Also, an auto-labeling pipeline using uncertainty improves performance on the 3D detection task.

**Strengths:**

- The paper's motivation is interesting, as most existing works only focus on improving model performance. Uncertainty estimation should play an important role in safety-critical applications, such as autonomous driving.

**Weaknesses:**

- The paper is a bit hard to follow. Most of the content describes different settings for different subtasks. However, the key contribution is hard to find, rather than applying uncertainty estimation to 3D object detection.
- Experimentally, conducting experiments simultaneously on camera-only, LiDAR-only, and fusion-based methods seems unnecessary. The input modality is unlikely to affect the main conclusion of the paper.
- Including the Waymo data set is only for constructing out-distribution data, which seems unnecessary (see Questions).
- The improvement is very marginal for the experimental results in section 4.5, which might also result from different random seeds.

**Questions:**

- In lines 366-368, the authors treat data from nuScenes as in-distribution data and data from Waymo as out-distribution data. This setting is quite doubtable since the number of cameras, camera settings, LiDAR settings, etc. are completely different. Differing from the paper cited in line 369 ([1]), which uses LiDAR-only detectors, this paper uses also camera-only and fusion-based methods. It is curious how the authors handle different numbers of cameras. As for the construction of OOD data, a straightforward method that comes to my mind is to treat daytime data from nuScenes as in-distribution data and nighttime data from nuScenes as out-distribution data. Why did the authors choose this setting mentioned in the paper?
- In line 459, the authors choose 15 as the number of potential missing objects. However, as discussed in the first paragraph in section 4.4, the uncertainty value identifies missing objects. It is double the choice of the number 15, and also, why not using the thresholds on uncertainty to estimate the locations of missing objects?
- The only experiment directly related to 3D object detection is section 4.5, where the experimental settings are a bit complicated. Does some kind of standard process exist in previous work? And why did the author choose these experimental settings?

[1] Wang, Yan, et al. "Train in germany, test in the usa: Making 3d object detectors generalize." Proceedings of the IEEE/CVF Conference on Computer Vision and Pattern Recognition. 2020.

---

> ### Author Response · Authors · 2024-11-18
> **Rebuttal by Authors**
>
> We appreciate the reviewer’s acknowledgment of the importance of uncertainty estimation in safety-critical applications like autonomous driving and their recognition of the interesting motivation behind our work. Below, we provide a detailed response to their feedback, addressing each concern while further clarifying our contributions and methodology.
>
> ---
>
> * **The paper is a bit hard to follow. Most of the content describes different settings for different subtasks. However, the key contribution is hard to find, rather than applying uncertainty estimation to 3D object detection.**
>
> We also recognize that the readability could be improved by better highlighting the key contribution and streamlining the presentation of tasks, which we will ensure in the final revised version of the paper.
>
> The key contribution lies in our novel uncertainty estimation framework, specifically designed for 3D object detection, which integrates evidential deep learning (EDL) with Bird’s Eye View (BEV) representations. This versatile framework enables computationally efficient (real-time) uncertainty quantification at multiple levels—scene, bounding box, and missed objects—while uniquely accounting for both classification and localization uncertainties simultaneously, a capability not addressed by other methods.
>
> The relabeling pipeline and the uncertainty use cases, such as OOD detection, bounding box quality assessment, and missed object identification, serve to demonstrate the effectiveness and practical applicability of our approach. These examples highlight how our method can improve the robustness and reliability of 3D object detection systems in real-world scenarios.
>
> * **Experimentally, conducting experiments simultaneously on camera-only, LiDAR-only, and fusion-based methods seems unnecessary. The input modality is unlikely to affect the main conclusion of the paper.**
>
> The inclusion of experiments on LiDAR-only and fusion-based methods serves to validate the generality of our uncertainty estimation framework across a variety of sensor setups commonly used in 3D object detection. While the input modality does not directly influence the core contribution of our method, using diverse scenarios ensures that the observed quality improvements are consistent across different setups. This consistency demonstrates that our improvements are not due to random variations from retraining the model but reflect a systematic and reliable enhancement, further reinforcing the robustness of our approach for real-world applications.
>
> * **Including the Waymo dataset is only for constructing out-distribution data, which seems unnecessary.**
>
> We used the Waymo dataset as out-of-distribution (OOD) data because a similar setup has been previously employed in studies such as [1] which highlights the practical importance of evaluating performance across different data distributions. That paper observed a significant drop in performance when models trained on one dataset were tested on a different one, such as NuScenes and Waymo, due to the OOD nature of the data.
>
> The nuScenes versus Waymo setup is not only well-established [1] but also aligns with standard protocols in domain adaptation [2, 3] and OOD generalization frameworks [4] in 3D object detection. Including Waymo in our evaluation follows this standard protocol, allowing us to demonstrate how our uncertainty estimation framework effectively identifies and handles OOD scenarios. This is a critical capability for real-world autonomous systems, where encountering OOD scenarios is common. We will revise the paper to emphasize these points and include additional references to clarify this choice.
>
> [1] Wang, Yan, et al. "Train in germany, test in the usa: Making 3d object detectors generalize." CVPR 2020.
>
> [2] Wang, Yan, et al. "Ssda3d: Semi-supervised domain adaptation for 3d object detection from point cloud." AAAI 2023
>
> [3] Tsai, Darren, et al. "Ms3d++: Ensemble of experts for multi-source unsupervised domain adaptation in 3d object detection." IEEE Transactions on Intelligent Vehicles 2024
>
> [4] Hegde, Deepti, et al. "Multimodal 3D Object Detection on Unseen Domains." arXiv 2024

---

> > ### Author Response · Authors · 2024-11-18
> >
> > * **The improvement is very marginal for the experimental results in section 4.5, which might also result from different random seeds.**
> >
> > We acknowledge the reviewer’s concern about the potential impact of random seed variability on the results. To address this, we have included variance estimates in Tables 1 and 2, which demonstrate that the improvements in Section 4.5 are consistent and statistically significant across multiple runs, computed as the mean and variance over three trainings with different seeds. These tables present results for two different setups, FocalFormer and DeformFormer (both LiDAR-based), showing that the observed differences in performance are robust and not attributable to random noise.
> >
> > **Table 1: Detection Performance for FocalFormer (LiDAR-based), 10k initial labelled scenes**
> >
> > | **Metric** | **Random** | **Pseudo** | **Ours** |
> > |------------|------------|------------|-----------|
> > | IoU        | 58.78 ± 0.07 | 59.38 ± 0.09 | 59.59 ± 0.04 |
> > | NDS        | 54.72 ± 0.05 | 55.64 ± 0.04 | 55.83 ± 0.06 |
> >
> > **Table 2: Detection Performance for DeformFormer (LiDAR-based), 10k initial labelled scenes**
> >
> > | **Metric** | **Random** | **Pseudo** | **Ours** |
> > |------------|------------|------------|-----------|
> > | IoU        | 58.42 ± 0.04 | 60.88 ± 0.05 | 61.09 ± 0.06 |
> > | NDS        | 54.30 ± 0.05 | 57.25 ± 0.07 | 57.66 ± 0.05 |
> >
> > The improvements in Section 4.5 consistently hold across multiple modalities and experimental setups, underscoring the robustness of our approach. Furthermore, these results are meaningful in the context of uncertainty-guided relabeling, which optimizes annotation efficiency and improves downstream performance.
> >
> > Additionally, on the three intermediary tasks of OOD detection, identifying bounding box errors, and undetected objects, our method outperforms baselines, often by margins exceeding 10%. These intermediary tasks validate the broader applicability and effectiveness of our framework, further supporting the utility of our uncertainty estimation approach.
> >
> > * **In lines 366-368, the authors treat data from nuScenes as in-distribution data and data from Waymo as out-distribution data. This setting is quite doubtable since the number of cameras, camera settings, LiDAR settings, etc. are completely different. Differing from the paper cited in line 369 ([1]), which uses LiDAR-only detectors, this paper uses also camera-only and fusion-based methods. It is curious how the authors handle different numbers of cameras. As for the construction of OOD data, a straightforward method that comes to my mind is to treat daytime data from nuScenes as in-distribution data and nighttime data from nuScenes as out-distribution data. Why did the authors choose this setting mentioned in the paper?**
> >
> > In our experiments, two of the setups we use—FocalFormer (L) and DeformFormer (L)—are LiDAR-based detectors, similar to the setup from [1]. The third setup, FocalFormer (L+C), is camera and LiDAR fusion, but we do not work with a camera-only setup.
> >
> > The question about handling datasets with different numbers of cameras is a valid concern. This issue can be addressed by either excluding excess cameras or replicating camera images to ensure compatibility across datasets, which is the approach we adopted. While these adjustments may introduce certain limitations, they enable us to evaluate our method across diverse configurations without undermining the validity of the experiments. However, it is important to note that most of our experimental setups did not face this "number of cameras" issue, as we primarily worked with LiDAR-based configurations.
> >
> > While a day/night split is an interesting approach to modeling OOD data, we use the nuScenes versus Waymo setup because it was used in [1] and is standard in the domain adaptation and generalization literature [2, 3, 4]. This approach allows us to evaluate our method in a realistic cross-dataset OOD scenario, which is common in autonomous driving applications.

---

> > > ### Author Response · Authors · 2024-11-18
> > >
> > > * **In line 459, the authors choose 15 as the number of potential missing objects. However, as discussed in the first paragraph in section 4.4, the uncertainty value identifies missing objects. It is double the choice of the number 15, and also, why not using the thresholds on uncertainty to estimate the locations of missing objects?**
> > >
> > > If we understood the question correctly, the reviewer is asking why we hard-set the number of missed object locations to 15. This choice was made with practical considerations in mind. In real-world scenarios, particularly in the context of autonomous driving, there needs to be a reasonable upper bound on the number of locations flagged for additional checks by human labelers to ensure the process remains efficient and feasible. On average, a typical 3D scene from the NuScenes dataset contains 10–20 objects, which makes 15 a reasonable choice as it aligns with the average scene complexity, avoiding overwhelming the labelers with excessive detections while still capturing most potential missed objects.
> > >
> > > If we were to rely solely on thresholding uncertainty values, it might result in an unmanageable number of flagged locations—for example, 100 or more—making the verification process impractical. By hard-setting the number to 15, we ensure a balance between identifying the most critical areas (high-uncertainty regions) and maintaining a practical workload for human verification. This decision aligns with the operational constraints of active learning and human-in-the-loop systems.
> > >
> > > * **The only experiment directly related to 3D object detection is section 4.5, where the experimental settings are a bit complicated. Does some kind of standard process exist in previous work? And why did the author choose these experimental settings?**
> > >
> > > While we understand the reviewer’s perspective, we respectfully disagree with the assertion that only the experiments in Section 4.5 are related to 3D object detection. All the experiments in our paper are designed to address critical aspects of 3D object detection:
> > >
> > > - The box-quality experiments in Section 4.3 directly assess the uncertainty of bounding boxes, which is fundamental to evaluating detection performance.
> > > - Section 4.4 tackles the task of missed object localization, a core challenge in 3D detection, ensuring that undetected objects are identified and addressed.
> > > - OOD scene detection, while broader in scope, is highly relevant to 3D object detection in autonomous systems, as it can significantly benefit downstream applications such as robust planning by identifying scenes where detection models might fail.
> > >
> > > Together, these experiments comprehensively demonstrate how our uncertainty estimation framework enhances various facets of 3D object detection and its associated tasks.
> > >
> > > Regarding Section 4.5 specifically, there is no strict standard process in prior work for combining uncertainty estimation with relabeling pipelines in 3D object detection. However, our approach builds on active learning principles widely used in related domains. The chosen experimental settings reflect practical scenarios where uncertainty estimates are used to prioritize high-uncertainty regions for relabeling, given a limited annotation budget. By evaluating uncertainty at multiple levels—scene, bounding box, and missed objects—we aimed to showcase how our framework can strategically guide relabeling while improving detection performance.
> > >
> > > Although the settings may appear complex, they were designed to replicate real-world constraints and emphasize the practical applicability of our method. This approach allows us to demonstrate not only the effectiveness of our uncertainty estimation but also its feasibility in operational scenarios.

---

> > > > ### Author Response · Authors · 2024-11-25
> > > >
> > > > Dear Reviewer 7H8K,
> > > >
> > > > Thank you for dedicating your time to reviewing our paper and providing valuable feedback.
> > > >
> > > > We have thoughtfully addressed each of your comments, offering detailed responses to clarify and resolve the concerns you raised. We hope our explanations have provided a clearer perspective on our work and its contributions.
> > > >
> > > > If you feel that we have adequately addressed your concerns, we would be grateful if you would consider reassessing your rating.
> > > >
> > > > We would be happy to clarify or elaborate on any part of our paper while the discussion period is still open.
> > > >
> > > > Thank you!

---

### Official Review · Reviewer_6GMQ · 2024-11-03

**Soundness:** 2
**Presentation:** 2
**Contribution:** 1
**Rating:** 3
**Confidence:** 5

**Summary:**

The authors propose a new method to quantify the uncertainty in 3D detection. By applying evidential learning on a bird's eye view feature map to predict the probability distribution of each Cell category, the uncertainty of the predicted bounding box can be obtained. The method is easy to implement and can be easily integrated into a 3D detector. The authors applied the uncertainty to several downstream tasks, comparing it with different baselines.

**Strengths:**

This paper investigates an important problem - uncertainty quantification for 3D detection results. The use of evidential learning enables a good efficiency in the inference stage, which meets the requirements of autonomous driving. A prior-based regularization term is added to the loss function to correct the Beta distribution, which is able to reduce the overconfidence of the model.

**Weaknesses:**

This paper lacks novelty, the method of quantifying uncertainty is not much different from that in paper [1]. Although the paper [1] is quantifying the uncertainty in the predictions of 2D detection models, quantifying uncertainty based on BEV features in this paper is not much different from that based on 2D image features. And also the quantification of uncertainty in the bounding box size is missing compared to the paper [1].
[1] Nallapareddy, Monish R., et al. "Evcenternet: Uncertainty estimation for object detection using evidential learning." 2023 IEEE/RSJ International Conference on Intelligent Robots and Systems (IROS). IEEE, 2023.

The experimental setup in this paper lacks transparency and lacks sufficient description for comparing methods. The 3D detection output is a bounding box defined by multiple parameters, each containing uncertainty. It is not clear in this paper which parameter uncertainties are included in the uncertainty generated by the comparison methods. Is it only the uncertainty of the center position as in the method proposed in this paper? The sampling-based approach to obtaining uncertainty in the predicted bounding boxes suffers from the problem of alignment, i.e., the outputs of the bounding boxes from different networks differ in number, location, and size, and the authors do not explain how it aggregates multiple predicted bounding boxes to derive the uncertainty.

The experiments presented in Table 2 should also be compared to the AUC metric of the original detector itself, i.e., the AUC of Confidence-IoU. In practice we usually use confidence to distinguish between correct and incorrect predictions. Whereas the experiments in Table 2 are designed to demonstrate that after quantifying uncertainty, using the uncertainty can better distinguish between correct and incorrect predictions. Therefore a comparison of confidence-IoU AUC is needed. Uncertainty needs to exceed the discriminative ability of the confidence to prove its suitability for this task.

In the missed object detection task proposed by the authors, the metric for judging the correctness of the predictions is not aligned with the previous section, i.e., IoU0.3. Using only the center distance to judge the correctness of the detection box is too lenient. In the authors' proposed method, all the cells with a prediction probability less than 5% are taken for missed object detection, which will make the network deal with a large number of background cells and will lead to a great sacrifice in computational efficiency. The low precision, recall, and F1-score in Table 3 illustrate that the improvement in the overall detection performance of the network is not significant. In practice, the simplest way to improve recall is to relax the confidence threshold, and whether the authors' proposed method is better than this simplest method needs to be proved experimentally.

The experimental results in Table 4 do not reflect that the method proposed by the authors provides significant enhancement. Moreover, the data in this table are not the average results of multiple experiments, which cannot provide a good convincing argument.

**Questions:**

In lines 427-428, why is the uncertainty of the predicted bounding box taken from the minimum value in the set?

The experiments in Table 2 present the quality of uncertainty quantification only from the perspective of AUC, can you add the results of the ECE metrics to provide proof from the calibration perspective?

---

> ### Author Response · Authors · 2024-11-20
> **Rebuttal by Authors**
>
> We sincerely thank the reviewer for their thoughtful evaluation and detailed feedback on our paper. We deeply appreciate the recognition of our work addressing the critical problem of uncertainty quantification in 3D object detection. We are particularly grateful for the acknowledgment of the efficiency of our evidential learning-based approach and the effectiveness of the prior-based regularization in mitigating model overconfidence. We have carefully considered all comments and suggestions and provide detailed responses to each concern below.
>
> ---
>
> * **This paper lacks novelty, the method of quantifying uncertainty is not much different from that in paper [1].**
>
> We appreciate the reviewer bringing up the comparison with the mentioned paper [1]. While we acknowledge certain similarities between the two approaches, we respectfully highlight several key distinctions that underscore the novelty and uniqueness of our work:
>
> - **Focus on Epistemic Uncertainty**: While [1] primarily integrates uncertainty into the inference pipeline to enhance detection quality, therefore focusing largely on aleatoric uncertainty, our approach prioritizes epistemic uncertainty. In safety-critical applications like autonomous driving, epistemic uncertainty is essential for understanding model reliability in out-of-distribution scenarios [2, 3], complex or unknown scene conditions [4], and situations with limited training data [5]. This emphasis on epistemic uncertainty not only distinguishes our contribution but also aligns naturally with active learning setups, where accurately quantifying uncertainty enables efficient selection of informative samples for labeling, ultimately improving model robustness and reliability.
>
> - **Different Distribution Formalization**: The mentioned paper adopts a Dirichlet distribution for modeling objectness heatmap predictions. In contrast, our approach employs a Beta distribution formulation, which is tailored to the requirements of modern 3D object detection architechtures. This distinction is important, as our formulation required deriving specific theoretical adjustments and practical modifications to integrate effectively within 3D detection pipelines. These differences make it unsuitable to directly borrow techniques from [1].
>
> - **Model Integration and Efficiency**: Our approach requires minimal architectural changes by adding a small component to the existing model, making it lightweight and easy to integrate into 3D detection pipelines. On the other hand, the mentioned approach [1] retrains the entire model to incorporate uncertainty estimation, requiring specialized losses and significantly modified architectures. Unlike this, our method is compatible with any model utilizing BEV features and works seamlessly with various training setups. This distinction underscores the efficiency of our method and highlights its practicality for large-scale applications, particularly in resource-constrained environments such as autonomous driving.
>
> - **3D vs. 2D Detection Context**: Quantifying uncertainty in 3D object detection poses fundamentally different challenges compared to 2D detection. While 2D detection also involves spatial relationships and object localization, it is limited to projecting objects onto a 2D plane, where the relationships between objects are inherently simpler. In contrast, 3D detection requires reasoning about spatial relationships in a three-dimensional environment, where factors like depth, scale, occlusion, and perspective significantly increase the complexity. Additionally, 3D detection often involves multi-modal inputs (e.g., LiDAR and camera), which must be fused effectively. Our method addresses these complexities by leveraging Bird's Eye View (BEV) features tailored for the 3D context, which, to the best of our knowledge, has not been done before in this way.
>
> We believe these points illustrate the distinct contributions of our work and its relevance to advancing uncertainty quantification for 3D object detection. We hope this clarifies the novelty and impact of our approach.

---

> > ### Author Response · Authors · 2024-11-20
> >
> > * **The experimental setup in this paper lacks transparency and lacks sufficient description for comparing methods. Is it only the uncertainty of the center position as in the method proposed in this paper? The sampling-based approach to obtaining uncertainty in the predicted bounding boxes suffers from the problem of alignment, i.e., the outputs of the bounding boxes from different networks differ in number, location, and size, and the authors do not explain how it aggregates multiple predicted bounding boxes to derive the uncertainty.**
> >
> > We thank the reviewer for raising concerns about the experimental setup and comparison methodology. We address these points below and will include this discussion in the revised paper:
> >
> > 1. **Uncertainty in Center Position**: Yes, we compute uncertainty only for the center position, consistent with how our method generates uncertainties. We did not compute uncertainty for other bounding box parameters such as size or rotation. This ensures a fair comparison across methods.
> >
> > 2. **Sampling-Based Methods**: For the five sampling-based methods, we followed standard practices to estimate uncertainty:
> >    - We used five samples at inference time, as recommended in [6].
> >    - As the uncertainty measure, we computed the entropy of the averaged distribution obtained from different samples, as described in [7].
> >    - For MC-Dropout, we set the dropout rate to 0.1 during inference, a value known to provide good results.
> >    - For Masksembles, we used a scale parameter of 2, as suggested in the original paper.
> >    - For Packed Ensembles, we followed the parameters outlined in the original publication.
> >
> > 3. **Alignment Issues in Sampling-Based Methods**: The issue of alignment (i.e., differing numbers, locations, and sizes of predicted bounding boxes across samples) can arise when applying sampling-based methods in a 3D detection context. However, there are no known adaptations of these methods specifically designed for the 3D detection setup. These methods are widely recognized as state-of-the-art for uncertainty quantification, and we evaluate them as they are commonly applied to ensure a fair comparison. This approach demonstrates the robustness and effectiveness of our method in the considered context.
> >
> > * **The experiments presented in Table 2 should also be compared to the AUC metric of the original detector itself, i.e., the AUC of Confidence-IoU.**
> >
> > We appreciate the reviewer’s suggestion regarding a comparison to the AUC metric of the original detector itself, specifically Confidence-IoU. The "Entropy" baseline already utilizes the confidence heatmap predictions of the original model, which are highly correlated with the confidence values for the final bounding boxes. In our observations, the Confidence-IoU metric performed similarly to the "Entropy" baseline, with no significant differences in results. Therefore, we decided to retain only the "Entropy" baseline in our evaluation to maintain consistency in the experimental setup across all methods.
> >
> > Additionally, as highlighted in the literature (e.g., [8]), confidence scores from neural networks are often poorly calibrated, making them less reliable as standalone uncertainty measures. This makes it rather expected that the sampling-based uncertainty baselines and our approach outperform confidence-based methods.

---

> > > ### Author Response · Authors · 2024-11-20
> > >
> > > * **In the missed object detection task proposed by the authors, the metric for judging the correctness of the predictions is not aligned with the previous section, i.e., IoU0.3. Using only the center distance to judge the correctness of the detection box is too lenient. In the authors' proposed method, all the cells with a prediction probability less than 5% are taken for missed object detection, which will make the network deal with a large number of background cells and will lead to a great sacrifice in computational efficiency. In practice, the simplest way to improve recall is to relax the confidence threshold, and whether the authors' proposed method is better than this simplest method needs to be proved experimentally.**
> > >
> > > Thank you for your comments. We would like to provide additional clarification regarding the evaluation setup to ensure everything is as clear as possible and will include this discussion in the revised paper for further clarity.
> > >
> > > 1. **Missed Object Detection vs. Erroneous Box Detection**: These two tasks are fundamentally different and should not be aligned. Erroneous box detection focuses on assessing the quality of the boxes predicted by the model, identifying cases where the bounding boxes are poorly localized. In contrast, missed object detection identifies objects in the 3D scene for which the model failed to produce any bounding box predictions. These tasks address entirely different issues and are evaluated using different metrics by design.
> > >
> > > 2. **Clarification on BEV Heatmap and <5% Threshold**: We understand that the role of the 5% threshold in our setup might require further explanation, so we’d like to clarify. Initially, we predict a BEV heatmap and identify cells with probabilities <5% as locations where we **potentially** could find missed objects. This step is important because the Transformer-based decoder in modern 3D detection models does not process these low-probability BEV cells, as it is constrained by a limited number of queries. As a result, the model does not produce predicted boxes with centers in these locations. Importantly, we do not classify all <5% locations as missed objects. Instead, these locations form a candidate set where we subsequently search for missed objects.
> > >
> > > 3. **Computational Efficiency**: This process does not lead to a sacrifice in computational efficiency. Operations such as masking, filtering, and convolutions used to identify potential missed object locations are lightweight and do not affect the overall efficiency of the model. The model's runtime remains comparable to the original architecture.
> > >
> > > 4. **Relaxing the Confidence Threshold**: Lowering the confidence threshold to improve recall would not be effective in this case, as modern Transformer-based detectors (such as DETR or FocalFormer) generate a predefined number of final detections regardless of confidence thresholds. Therefore, our approach, which explicitly identifies low-confidence locations and searches for potential missed objects within them, is a more principled way to address the missed object detection task.

---

> > > > ### Author Response · Authors · 2024-11-20
> > > >
> > > > * **The low precision, recall, and F1-score in Table 3 illustrate that the improvement in the overall detection performance of the network is not significant.**
> > > >
> > > > We thank the reviewer for their comments and would like to provide a detailed clarification regarding the evaluation methodology for precision, recall, and F1-score as presented in Table 3. These metrics, in this case, are not evaluated in the same manner as in standard classification or detection tasks. Instead, they are specifically designed for the missed object detection task, which is fundamentally different.
> > > >
> > > > In this task, we predict a predefined number of locations (namely, 15) where we consider missed objects might be present. As mentioned previously, simply lowering the confidence threshold to improve recall is not applicable in this context because Transformer-based detectors (such as DETR or FocalFormer) produce a fixed number of detections, and thresholding does not control the recall of the detections, since all of the detections are used for the evaluation.
> > > >
> > > > * **The experimental results in Table 4 do not reflect that the method proposed by the authors provides significant enhancement.**
> > > >
> > > > We acknowledge the reviewer’s concern regarding the significance of the enhancements presented in Table 4. To clarify, we have provided variance estimates in our experiments to ensure that the reported improvements are robust and statistically significant. The results in Tables 1 and 2, which represent two different setups (FocalFormer and DeformFormer, both LiDAR-based), are computed as the mean and variance over three trainings with different random seeds. These results demonstrate that the observed performance gains are consistent across multiple runs and are not attributable to random noise.
> > > >
> > > > **Table 1: Detection Performance for FocalFormer (LiDAR-based), 10k initial labelled scenes**
> > > >
> > > > | **Metric** | **Random** | **Pseudo** | **Ours** |
> > > > |------------|------------|------------|-----------|
> > > > | IoU        | 58.78 ± 0.07 | 59.38 ± 0.09 | 59.59 ± 0.04 |
> > > > | NDS        | 54.72 ± 0.05 | 55.64 ± 0.04 | 55.83 ± 0.06 |
> > > >
> > > > **Table 2: Detection Performance for DeformFormer (LiDAR-based), 10k initial labelled scenes**
> > > >
> > > > | **Metric** | **Random** | **Pseudo** | **Ours** |
> > > > |------------|------------|------------|-----------|
> > > > | IoU        | 58.42 ± 0.04 | 60.88 ± 0.05 | 61.09 ± 0.06 |
> > > > | NDS        | 54.30 ± 0.05 | 57.25 ± 0.07 | 57.66 ± 0.05 |
> > > >
> > > > These results emphasize the robustness and consistency of our approach across different architectures and experimental setups. The improvements are meaningful within the context of uncertainty-guided relabeling, where optimizing annotation efficiency and improving downstream performance are critical goals.
> > > >
> > > > Furthermore, on the three intermediary tasks—OOD detection, identifying bounding box errors, and detecting undetected objects—our method outperforms baselines by significant margins, often exceeding 10%. These intermediary tasks validate the broader applicability and effectiveness of our framework, providing strong evidence for the utility of our uncertainty estimation method in real-world scenarios.
> > > >
> > > > **Evaluation Setup**: We evaluate precision, recall, and F1-score in the context of identifying missed objects as follows:
> > > > 1. **Missed Object Definition**: For a given scene, the ground truth boxes that have a low IoU with all predicted boxes are considered missed objects, as they are not covered by the model's predictions.
> > > >
> > > > 2. **Predicted Locations**: Our method identifies potential locations for missed objects, based on low-confidence regions in the BEV heatmap, as previously explained.
> > > >
> > > > 3. **True Positives and False Negatives**:
> > > >    - If a proposed location for a missed object is within a defined vicinity of a ground truth missed object (e.g., 2 or 4 meters), it is considered a true positive.
> > > >    - If there is no predicted location within the vicinity of an actual missed object, it is considered a false negative.
> > > >
> > > > Precision, Recall, and F1-Score metrics are then calculated based on the true positives and false negatives, treating the missed object detection task as a classification problem.
> > > >
> > > > **Relevance of Results**: The recall of approximately 10% achieved by our model indicates that it can successfully localize 10% of the objects missed by the original model. These are objects that cannot be easily recovered through simple methods such as adjusting soft thresholding due to the predefined detection setup of Transformer-based models. This demonstrates that our approach provides a meaningful improvement in identifying objects that the model would otherwise fail to detect.

---

> > > > > ### Author Response · Authors · 2024-11-20
> > > > >
> > > > > * **In lines 427-428, why is the uncertainty of the predicted bounding box taken from the minimum value in the set?**
> > > > >
> > > > > The motivation for using the minimum value of the uncertainties within the bounding box stems from our empirical observations. We experimented with different quantiles of the uncertainties, including the mean, median, and lower quantiles, and found that lower quantiles yielded better results. Based on these observations, we chose the minimum as our final uncertainty measure.
> > > > >
> > > > > Our explanation for this behavior is that higher quantiles (e.g., max) can be less robust because a single BEV cell with high uncertainty within the box could mark the entire box as uncertain, even if the majority of the cells are confident. By contrast, taking a more conservative approach with smaller quantiles, such as the minimum, ensures that the uncertainty measure reflects a more robust assessment of the overall box uncertainty.
> > > > >
> > > > > * **The experiments in Table 2 present the quality of uncertainty quantification only from the perspective of AUC, can you add the results of the ECE metrics to provide proof from the calibration perspective?**
> > > > >
> > > > > Calibration metrics, such as ECE, are indeed valuable for assessing uncertainty quantification. However, calibration can often be improved independently of the core uncertainty estimation method by applying post-hoc techniques like temperature scaling or other calibration methods. As a result, while ECE provides insight into calibration, it does not fully capture the inherent quality of the uncertainty estimation method itself.
> > > > >
> > > > > In Table 2, we focused on AUC metrics to evaluate the ranking quality of uncertainties and their ability to distinguish between high and low confidence predictions. This approach aligns with the practical needs of safety-critical applications like autonomous driving, where effective uncertainty ranking is more immediately actionable.
> > > > >
> > > > > We recognize the importance of calibration and will consider exploring metrics like ECE in future work to provide a broader perspective on uncertainty evaluation.
> > > > >
> > > > > [1] Nallapareddy, Monish R., et al. "Evcenternet: Uncertainty estimation for object detection using evidential learning." IROS 2023.
> > > > >
> > > > > [2] Michelmore, Rhiannon, et al. "Uncertainty quantification with statistical guarantees in end-to-end autonomous driving control." ICRA 2020.
> > > > >
> > > > > [3] Filos, Angelos, et al. "Can autonomous vehicles identify, recover from, and adapt to distribution shifts?." ICML 2020.
> > > > >
> > > > > [4] Kendall, Alex, and Yarin Gal. "What uncertainties do we need in bayesian deep learning for computer vision?." NeurIPS 2017.
> > > > >
> > > > > [5] McAllister, Rowan, et al. "Concrete problems for autonomous vehicle safety: Advantages of bayesian deep learning." IJCAI 2017.
> > > > >
> > > > > [6] Wen, Yeming, et al. "Batchensemble: an alternative approach to efficient ensemble and lifelong learning." ICLR 2020.
> > > > >
> > > > > [7] Durasov, Nikita, et al. "Masksembles for uncertainty estimation." CVPR 2021.
> > > > >
> > > > > [8] Guo, Chuan, et al. "On calibration of modern neural networks." ICML 2017.

---

> > > > > > ### Author Response · Authors · 2024-11-25
> > > > > >
> > > > > > Dear Reviewer 6GMQ,
> > > > > >
> > > > > > Thank you for dedicating your time to reviewing our paper and providing valuable feedback.
> > > > > >
> > > > > > We have thoughtfully addressed each of your comments, offering detailed responses to clarify and resolve the concerns you raised. We hope our explanations have provided a clearer perspective on our work and its contributions.
> > > > > >
> > > > > > If you feel that we have adequately addressed your concerns, we would be grateful if you would consider reassessing your rating.
> > > > > >
> > > > > > We would be happy to clarify or elaborate on any part of our paper while the discussion period is still open.
> > > > > >
> > > > > > Thank you!

---

> > > > > > > ### Comment · Reviewer_6GMQ · 2024-11-28
> > > > > > >
> > > > > > > Thank all authors that provided further rebuttal information. This paper successfully applies evidence learning to uncertainty quantification for 3D detection, which significantly improves OOD detection compared to other methods and can also improve the prediction of poorly localized bounding boxes. Defining a bounding box requires multiple parameters such as center position, size, and rotation angle; I suggest the authors extend their method to measure the uncertainty of each parameter of the bounding box as the sampling-based method is able to quantify all of them. A comparison based on the uncertainty of all bounding box parameters would be more fair. I appreciate that the authors expanded the use of uncertainty in 3D detection by applying uncertainty to missed object detection, but the presented results do not indicate that the method can significantly improve the overall performance of 3D detection. Using the author's labeling strategy to generate pseudo-labels also does not provide a noticeable improvement compared to other models, and considering the potential noise propagation problem that leads to a decrease in the model's generalization ability, the application of this method to safety-critical 3D detection models is still not sufficiently convincing.

---

### Official Review · Reviewer_Ne6n · 2024-11-04

**Soundness:** 2
**Presentation:** 3
**Contribution:** 3
**Rating:** 8
**Confidence:** 3

**Summary:**

This paper tackles the crucial issue of uncertainty estimation in 3D object detection for autonomous driving. It applies Evidential Deep Learning (EDL) to Bird’s Eye View (BEV) representations to provide efficient uncertainty estimates. Experiments show that this approach improves detection reliability, especially in challenging cases like out-of-distribution scenes and missed objects.

**Strengths:**

S1：The theoretical foundation is solid, with a novel adaptation of Evidential Deep Learning to 3D detection.

S2：Experimental results are promising and show meaningful improvements in several key tasks for autonomous driving.

S3：The approach appears innovative and applicable to real-world challenges in autonomous driving.

**Weaknesses:**

W1: The method mainly focuses on providing uncertainty estimates without apparent direct improvement in detection accuracy itself.

W2: While experiments cover LiDAR and LiDAR+Camera setups, a significant portion of current detection models are Camera-only; hence, the framework lacks some supportion in experiment to demenstrate the generality in this area.

W3: The paper claims efficiency advantages over traditional methods, suggesting suitability for real-time applications like driving; however, it does not provide a direct comparison in terms of computational efficiency. It remains unclear how much this added uncertainty estimation slows down the original model.

**Questions:**

Q1: It's well understood that traditional methods like MC Dropout and Deep Ensembles have high computational costs due to repeated inference, which is why they’re increasingly less popular in uncertainty quantification for autonomous driving. Given this shift, the question is: compared to existing UQ directly modeling methods, such as [1], does your evidential learning approach offer any distinct advantages?

[1] Uncertainty Quantification of Collaborative Detection for Self-Driving

Q2: Is it possible to extend your approach to other BEV-based perception tasks, such as occupancy prediction?

---

> ### Author Response · Authors · 2024-11-23
> **Rebuttal by Authors**
>
> We sincerely appreciate the reviewer’s thoughtful feedback on our work. We are especially grateful for the recognition of the theoretical foundation of our method, the meaningful experimental results, and the innovation and applicability of our approach to real-world challenges in autonomous driving. Below, we carefully address all the concerns and suggestions raised.
>
> ---
>
> * **The method mainly focuses on providing uncertainty estimates without apparent direct improvement in detection accuracy itself.**
>
> Indeed, the primary goal of our method is not to directly improve detection accuracy but to provide reliable uncertainty estimates while also maintaining the accuracy of the considered model. Comparisons between the original model and our method confirm that the accuracy remains consistent, ensuring that our approach adds uncertainty estimation capabilities without compromising detection performance. However, our active learning setup demonstrates how uncertainty estimation can make the labeling process for self-driving applications more efficient. By strategically prioritizing high-uncertainty regions for relabeling, our approach indirectly improves detection accuracy by ensuring higher-quality training data.
>
> * **While experiments cover LiDAR and LiDAR+Camera setups, a significant portion of current detection models are Camera-only; hence, the framework lacks some supportion in experiment to demenstrate the generality in this area.**
>
> While it is true that we focused more on setups that include LiDAR information in inference, our method operates at the BEV level, which is agnostic to the type of inputs used to generate predictions. The choice of input modalities primarily affects the quality of the BEV features, which is beyond the scope of our work, as we directly use the BEV representations provided by the backbone.
>
> That said, we agree that including experiments with camera-only setups would further strengthen the demonstration of our method’s generality. However, conceptually, such experiments would not alter the main conclusions of the paper, as also pointed out by reviewer #7H8K. We will consider this extension in future work to provide a more comprehensive evaluation.
>
> * **The paper claims efficiency advantages over traditional methods, suggesting suitability for real-time applications like driving; however, it does not provide a direct comparison in terms of computational efficiency. It remains unclear how much this added uncertainty estimation slows down the original model.**
>
> Our method introduces minimal computational overhead, as it only modifies a small part of the model: the heatmap prediction head. This component consists of a few lightweight convolutional layers applied directly on top of the BEV features. As a result, the additional computational cost is negligible compared to the overall model inference time, amounting to roughly 1% of the original inference time.
>
> In contrast, standard sampling-based approaches require multiple forward passes (N times the inference time), making our method significantly more efficient. This negligible overhead underscores the suitability of our approach for real-time applications like autonomous driving.
>
> * **It's well understood that traditional methods like MC Dropout and Deep Ensembles have high computational costs due to repeated inference, which is why they’re increasingly less popular in uncertainty quantification for autonomous driving. Given this shift, the question is: compared to existing UQ directly modeling methods, such as [1], does your evidential learning approach offer any distinct advantages?**
>
> Evidential learning has been gaining popularity recently due to its ability to efficiently estimate uncertainty while providing high-quality epistemic uncertainty estimates—a notable weak spot for many existing sampling-free methods. Unlike sampling-based approaches, which are well-suited for estimating epistemic uncertainty but come with prohibitive computational costs, evidential learning offers an appealing trade-off by combining efficiency with robust epistemic uncertainty estimation.
>
> Regarding [1], while the proposed method falls under the category of sampling-free uncertainty estimation, it directly predicts predictive variance. This approach is known to poorly represent epistemic uncertainty, as it often conflates aleatoric and epistemic uncertainty, leading to potential limitations in its applicability. Evidential learning, in contrast, provides a principled way to model epistemic uncertainty explicitly, making it a compelling alternative in scenarios requiring efficient and reliable uncertainty quantification.
>
> [1] Su, Sanbao, et al. "Uncertainty quantification of collaborative detection for self-driving." ICRA 2023.

---

> > ### Author Response · Authors · 2024-11-23
> >
> > * **Is it possible to extend your approach to other BEV-based perception tasks, such as occupancy prediction?**
> >
> > Yes, our approach can be naturally extended to other BEV-based perception tasks, such as occupancy prediction. Operating at the BEV level, our method is task-agnostic and applicable to scenarios like estimating occupancy probabilities, refining semantic mapping, predicting motion with uncertainty, or delineating road structures. This flexibility highlights its broad applicability to various autonomous driving and robotics tasks.

---

> > > ### Author Response · Authors · 2024-11-28
> > >
> > > Dear Reviewer Ne6n,
> > >
> > > Thank you for dedicating your time to reviewing our paper and providing valuable feedback.
> > >
> > > We have thoughtfully addressed each of your comments, offering detailed responses to clarify and resolve the concerns you raised. We hope our explanations have provided a clearer perspective on our work and its contributions.
> > >
> > > We would be happy to clarify or elaborate on any part of our paper while the discussion period is still open.
> > >
> > > Thank you!

---

> > > > ### Comment · Reviewer_Ne6n · 2024-12-01
> > > >
> > > > Thank you to the authors for the further clarifications and detailed responses. Your explanations were clear and comprehensive. I have no additional questions.

---

### Official Review · Reviewer_n1q7 · 2024-11-04

**Soundness:** 4
**Presentation:** 4
**Contribution:** 2
**Rating:** 5
**Confidence:** 5

**Summary:**

The paper discusses the importance of 3D object detection from LiDAR and camera images and how deep neural networks, while excellent at detections, struggle with assessing reliability. The authors propose a computationally efficient framework motivated by Evidential Deep Learning (EDL), which outputs parameters for a distribution over class probabilities, referred to as a second-order distribution. This allows the model to estimate both the probability of an object's presence and the associated uncertainty.

The authors extend this framework to 3D object detection by adapting the model architecture and loss function to generate uncertainty estimates. They replace the standard heatmap head with an EDL head, which predicts both object presence probabilities and uncertainty. The loss function is designed to address class imbalance and focuses on harder, misclassified examples during training.

The authors conduct the experiments on three downstream applications: detecting out-of-distribution scenes, assessing the quality of predicted bounding boxes, and identifying missed objects. The authors also integrate these tasks into an auto-labeling pipeline where uncertainty estimates are used to verify label correctness before training a second model, resulting in improvements in mAP and NDS.

**Strengths:**

The paper is well-organized and clearly written. The introduction effectively outlines the problem and the motivation behind the work. The methodology section is detailed, providing clear explanations of the EDL framework, the model architecture, and the loss function. The results are presented in a manner that is easy to understand, with comprehensive tables and figures that clearly illustrate the performance improvements.

**Weaknesses:**

The proposed methodology integrates an advanced annotation phase to augment object detection capabilities. However, it does not adequately address the limitations of traditional approaches, necessitating continued extensive re-annotation efforts without a significant leap in detection efficiency. Moreover, the key to enhancing performance in real-time robotics applications, particularly in autonomous driving, lies in the effective management of uncertainty during the detection phase, rather than the re-annotation process. This is crucial for applications such as motion planning, which requires sophisticated algorithms capable of adapting to dynamic and unpredictable environments.

The authors have conducted three downstream experiments to validate their approach: detecting out-of-distribution scenes, assessing the quality of predicted bounding boxes, and identifying missed objects. These experiments are essential for understanding the robustness of the detection system. However, the paper does not provide clear evidence on how the uncertainty estimation directly benefits real autonomous driving applications, such as motion planning. Therefore, while the experiments conducted are valuable, there is a need to demonstrate how uncertainty estimation translates into tangible benefits for autonomous driving, particularly in the context of motion planning and real-time decision-making.

======

**Post-rebuttal**:

- High-Quality Annotation's Essential Role in Object Detection: The precision of object detection models is fundamentally contingent upon the quality of the annotations they are trained on. This manuscript innovatively incorporates uncertainty estimation to curtail the number of objects necessitating re-annotation, thereby optimizing the annotation process. However, it is crucial to acknowledge that while uncertainty estimation aids in this optimization, the ultimate accuracy of object detection models is still predicated on the fidelity of the re-annotation efforts.

- Uncertainty Estimation's Dual Role: The authors have overlooked a pivotal aspect of uncertainty estimation—its role in risk management. This component is particularly significant for autonomous driving systems, where it can significantly bolster safety by providing a measure of confidence in detection outcomes. The initial submission underplays this aspect, which diminishes the perceived value of uncertainty estimation in practical applications.

- Enhancing the Contribution of Uncertainty Estimation: To rectify this oversight, I suggest that the authors expand on how uncertainty estimation can be integrated into risk assessment protocols for autonomous vehicles. By doing so, they can highlight the dual benefits of their approach: not only does it streamline the annotation process, but it also enhances the reliability and safety of autonomous systems through robust risk management.

**Questions:**

1. Please provide details in the auto-label pipeline: no reference and description referring to the auto-label pipeline.

---

> ### Author Response · Authors · 2024-11-18
> **Rebuttal by Authors**
>
> # Review #n1q7
>
> We are grateful to the reviewer for their detailed review and valuable suggestions regarding our paper. We also deeply appreciate their recognition of of the clarity in our writing and the thoroughness of our experiments, while also considering strong aspects of our method. We have carefully addressed all of the concerns and suggestions raised, as discussed below.
>
> ---
>
> * **The proposed method does not adequately address the limitations of traditional approaches, necessitating continued extensive re-annotation efforts without a significant leap in detection efficiency.**
>
> We appreciate the reviewer’s observation regarding annotation requirements. Indeed, our method does not eliminate the need for labeled data, and we did not claim it would. Effective training, especially for complex tasks like 3D object detection, inherently benefits from additional labeled data, and our approach aligns with this understanding.
>
> The core contribution of our paper is an improved uncertainty estimation method that outperforms traditional approaches in efficiency and effectiveness. Specifically, we demonstrate its advantages in an active learning setup, where our method allows for more strategic data selection, thereby reducing the volume of new annotations needed compared to existing uncertainty methods. In other words, while labeling remains essential, our approach allows for a more efficient, targeted annotation process,
>
> * **The key to enhancing performance in real-time robotics applications, particularly in autonomous driving, lies in the effective management of uncertainty during the detection phase, rather than the re-annotation process.**
>
> Our approach is designed to effectively manage uncertainty during inference. We will clarify the wide applicability of our framework in the revised paper to ensure this point is more evident. While our final experiment focuses on a re-annotation task, our initial experiments on OOD detection, bounding box quality assessment, and missed object identification can also directly be used to drive real-time processes in robotics and autonomous driving. For example, our method effectively flags high-uncertainty scenes, enabling autonomous systems to hand control back to human drivers—a feature that directly enhances operational safety. These showcases underscore our approach’s value in dynamically managing uncertainty during detection, making it highly applicable for real-time settings.
>
> * **The paper does not provide clear evidence on how the uncertainty estimation directly benefits real autonomous driving applications, such as motion planning.**
>
> We will revise the paper to better emphasize how our method effectively benefits real-world applications, including motion planning, as mentioned by the reviewer. Our paper demonstrates how improved uncertainty estimation directly benefits real-world applications like motion planning. By identifying uncertain regions or detections during inference, our approach enables adaptive responses, such as human intervention or planning adjustments in high-risk situations. This supports safer and more informed decision-making, which is critical for autonomous driving. While enhancing the planner itself is outside the scope of this work, as it is a distinct and complex task, our method provides valuable uncertainty insights that can improve the safety and reliability of the planning process through effective uncertainty management.

---

> > ### Author Response · Authors · 2024-11-18
> >
> > * **Please provide details in the auto-label pipeline: no reference and description referring to the auto-label pipeline**
> >
> >
> > We will add further details below in our response and include them in the revised version of the paper to ensure clarity and address this concern. In Section 4.5, we describe how our framework integrates uncertainty estimation into the auto-labeling process. Here’s a more detailed explanation of the pipeline:
> >
> > Our approach begins by training a 3D object detector on a subset of randomly selected scenes from the NuScenes training dataset (10,000 and 20,000 scenes in our experiments). This trained detector generates pseudo-labels for the remaining unlabeled scenes in the NuScenes training set. We then apply our uncertainty estimation framework to analyze these pseudo-labels at multiple levels:
> >
> > - **Scene-Level Uncertainty**: We compute the average uncertainty across the entire scene to identify OOD instances or scenes with unreliable data (Section 4.2).
> > - **Bounding Box-Level Uncertainty**: Predicted bounding boxes are evaluated for uncertainty to identify likely errors that may require correction or verification (Section 4.3).
> > - **Missed Object Detection**: Regions with high uncertainty, where objects are potentially missed, are flagged to address false negatives (Section 4.4).
> >
> > The pipeline proceeds as follows:
> > 1. **Scene-Level Relabeling**: After calculating scene-level uncertainties, we select the scenes with the highest uncertainty values. To emulate the relabeling process, we assume that all ground truth objects in these scenes are re-verified/re-labelled, consuming the scene-level uncertainty budget. For instance, if a scene contains 10 ground truth objects, 10 boxes are deducted from the allocated scene-level budget of 10,000 boxes.
> > 2. **Bounding Box-Level Verification**: After exhausting the scene-level budget, we focus on the bounding boxes with the highest uncertainty. These are reviewed and corrected within the box-level budget of 10,000 boxes.
> > 3. **Missed Object Detection**: Lastly, we use the same 10,000 boxes budget to search for and correct missed detections, targeting regions with high uncertainty.
> >
> > This structured approach ensures efficient allocation of labeling resources while maximizing the quality of pseudo-labels. The retrained model using these enhanced labels demonstrates substantial improvements, validating the effectiveness of this pipeline.
> >
> > **Dataset and Budget Details**
> >
> > **Table 1: Data Setup**
> >
> >
> > | **Method**                  | **Initial Number of Scenes** | **Unlabeled Scenes Pool Size** | **Description**                                                                 |
> > |--------------------------------|-----------------------------|---------------------------------|---------------------------------------------------------------------------------|
> > | Random                         | N thousand                  | 30-N thousand                  | Train the 3D object detector on randomly selected scenes, no pseudo-labeling.  |
> > | Pseudo-Labeling                | N thousand                  | 30-N thousand                  | Generate pseudo-labels for the unlabeled pool and retrain the model on 30 thousand scenes. |
> > | Ours (Pseudo-Labeling + Verification) | N - 1 thousand           | 31-N thousand                  | Correct pseudo-labels for 1 thousand scenes (or 30 thousand boxes in total) through uncertainty-guided verification. |
> >
> > ---
> >
> > **Table 2: Uncertainty Methods and Budgets**
> >
> > | **Uncertainty Use Case**     | **Budget (Boxes)** | **Description**                                        |
> > |-----------------------------|--------------------|--------------------------------------------------------|
> > | Scene-Level Uncertainty     | 10,000            | Relabeling all objects in the most uncertain scenes.   |
> > | Bounding Box-Level Uncertainty | 10,000        | Fixing or verifying bounding boxes with highest uncertainty. |
> > | Missed Object Detection     | 10,000            | Correcting missed detections in high-uncertainty regions. |
> >
> > This table outlines the data setup and box budgets used in our experiments, detailing how resources were allocated across different relabeling and uncertainty-based methods within our pipeline.

---

> > ### Comment · Reviewer_n1q7 · 2024-11-19
> >
> > **Overview**: I appreciate the rebuttal provided by the authors. I am gratified that we have reached a consensus on the importance of uncertainty detection in risk management. However, it is important to note that consensus has not yet been achieved on the re-annotation process, which remains a contentious issue. The authors have not presented direct evidence to support the claim that their proposed method significantly enhances the efficiency and effectiveness of re-annotation. Furthermore, the performance improvements cited in object detection are still contingent upon the quality of re-annotation.
> >
> > - Can you provide the evidence to support the proposed method for reducing the volume of re-annotation objects compared to existing methods in active learning?
> >
> > - It is appreciated to see the application in risk management.
> >
> > - Again, it is appreciated to see the application in risk management, rather than re-annotation.

---

> ### Comment · Reviewer_n1q7 · 2024-11-19
>
> Thank you for providing the details of auto-label pipeline.

---

> > ### Author Response · Authors · 2024-11-23
> > **Authors Response to Reviewer n1q7**
> >
> > * **Can you provide the evidence to support the proposed method for reducing the volume of re-annotation objects compared to existing methods in active learning?**
> >
> > To support the claim that our proposed method reduces the volume of re-annotation objects compared to existing methods in active learning, we provide evidence in Tables 1 and 2 below. These tables demonstrate that, for a given target IoU, our method consistently requires fewer labeled samples while achieving better NDS values compared to Random and Pseudo-Labeling baselines. These results, derived using interpolation, highlight the efficiency of our uncertainty-guided approach in optimizing the relabeling process.
> >
> > For instance, in Table 1, our method saves 2,200 labeled samples compared to Random, reducing labeling costs by \\$2,200 (\\$1 per scene, a realistic price) or 12.5%. To put it another way, this demonstrates the scalability of our approach, as it allows more labeling to be done within the same budget, maximizing the efficiency of available resources. Similarly, in Table 2, our method saves 4,400 labeled samples, corresponding to $4,400 or 30% in cost reduction. These results span significantly different setups (LiDAR-only in Table 1 and Camera + LiDAR in Table 2) and demonstrate the generality of our method. For both configurations, our approach achieves superior performance with fewer re-annotation objects, validating its efficiency and effectiveness in reducing labeling costs while improving detection quality.
> >
> > **Table 1. Labeled samples and NDS for IoU = 0.62 (FocalFormer, LiDAR). Our method reduces labeling needs while improving NDS.**
> > | **Base Model** | **Random** | **Ps.-Lab.** | **Ours** |
> > |----------------|------------|--------------|----------|
> > | # Samples      | 17,600     | 16,800       | 15,400   |
> > | NDS            | 0.5651     | 0.5699       | 0.5817   |
> >
> > **Table 2. Labeled samples and NDS for IoU = 0.68 (FocalFormer, Camera + LiDAR). Our method achieves better NDS with fewer labels.**
> > | **Base Model** | **Random** | **Ps.-Lab.** | **Ours** |
> > |----------------|------------|--------------|----------|
> > | # Samples      | 14,500     | 11,600       | 10,100   |
> > | NDS            | 0.5965     | 0.5979       | 0.6004   |
> >
> > * **It is appreciated to see the application in risk management.**
> >
> > While we agree that risk management in motion planning is important, we would like to emphasize that our primary focus is to demonstrate (i) the general ability to obtain uncertainty estimates and (ii) their application to the auto-labeling pipeline. We note that auto-labeling is a significant downstream use case of uncertainty estimation in autonomous vehicles, as demonstrated in prior literature [1-3].
> >
> > Nonetheless, to address the reviewer’s request and further explore the potential applications of our method, we conducted additional experiments to illustrate its relevance to risk management tasks. For example, our evaluation revealed that in roughly 24% of scenes, the 3D detection model misses at least one object. In a pipeline where perception outputs (from the 3D detector) are passed to a planner model responsible for generating a trajectory, this can lead to catastrophic outcomes in the worst-case scenario. Specifically, the planner may generate a trajectory through an undetected object, posing significant safety risks.
> >
> > To mitigate such risks, our uncertainty estimation framework can act as a risk management layer. By leveraging uncertainty, we can identify high-risk situations where objects are likely missed, enabling fallback mechanisms, such as handing control back to the driver. Without any uncertainty-based risk management, the vehicle would crash into undetected objects in all such cases. However, using our approach, similar to the method described in Section 4.4, we can effectively detect 10–20% of these high-risk situations (e.g., ~15% for the LiDAR-based FocalFormer model and ~11% for the DeformFormer model, as shown in Table 3). This capability significantly enhances safety, reducing the likelihood of undetected object-related incidents and ensuring more robust planning and decision-making in real-world scenarios.
> >
> > **Table 3. Recall (%) of high-risk situations detected using our uncertainty estimation framework.**
> > |  | **FocalFormer (L)** | **DeformFormer (L)** |
> > |----------------|------------|--------------|
> > | Recall, %      |   14.9   | 11.2       |
> >
> > [1] Wang, Yan, et al. "Ssda3d: Semi-supervised domain adaptation for 3d object detection from point cloud." AAAI 2023
> >
> > [2] Tsai, Darren, et al. "Ms3d++: Ensemble of experts for multi-source unsupervised domain adaptation in 3d object detection." IEEE Transactions on Intelligent Vehicles 2024
> >
> > [3] Hegde, Deepti, et al. "Multimodal 3D Object Detection on Unseen Domains." arXiv 2024

---

> > > ### Author Response · Authors · 2024-11-25
> > >
> > > Dear Reviewer n1q7,
> > >
> > > Thank you for dedicating your time to reviewing our paper and providing valuable feedback.
> > >
> > > We have thoughtfully addressed each of your comments, offering detailed responses to clarify and resolve the concerns you raised. We hope our explanations have provided a clearer perspective on our work and its contributions.
> > >
> > > If you feel that we have adequately addressed your concerns, we would be grateful if you would consider reassessing your rating.
> > >
> > > We would be happy to clarify or elaborate on any part of our paper while the discussion period is still open.
> > >
> > > Thank you!

---

### Meta-Review · Area_Chair_jJ9J · 2024-12-23

**Metareview:**

The paper proposes an evidential learning loss in BEV representations for quantifying uncertainty in 3D object detection. This is an important problem in applications such as autonomous driving, where the proposed method demonstrates benefits such as detecting OOD scenarios, false negatives and incorrect localizations. A key limitation of the paper is that the connection to improved detection accuracy and reliability in autonomous driving is only indirect, through a re-annotation process based on the uncertainty quantification. But for the latter, the obtained improvements in the auto-labeling application are also not large. Overall, the paper addresses an important problem and presents new ideas, but needs further improvements to justify itself conceptually and empirically. Thus, it is not recommended for acceptance, but the authors are encouraged to incorporate review feedbacks to resubmit to a future venue.

**Additional Comments On Reviewer Discussion:**

Reviewer n1q7 surmises that the paper essentially optimizes the annotation effort without directly affecting detection accuracy. While several details on the auto-labeling procedure are provided by the rebuttal, this fundamental challenge remains in the opinion of the reviewer. Ne6n shares some of the above concerns, but is satisfied by the author rebuttal and recommends acceptance. Reviewer 7H8K does not support acceptance based on the presentation and experimental choices. Reviewer 6GMQ questions similarity to prior works and while this is partially addressed by the rebuttal, remains unconvinced on the appropriateness of the uncertainty computation for the specific case of 3D bounding boxes. Overall, the majority of reviewers appreciate the contributions of the paper, but suggest several improvements for a revision towards a future submission.

---

### Decision · Program_Chairs · 2025-01-22

Reject